# MxB inhibits long interspersed element type 1 retrotransposition

Yu Huang[1]☯, Fengwen Xu[1]☯, Shan Mei[1], Xiaoman Liu[1], Fei Zhao[1], Liang Wei[1], Zhangling Fan[1], Yamei Hu[1], Liming Wang[2], Bin Ai[2], Shan Cen[3], Chen Liang[4]*, Fei Guo[1]*

1 NHC Key Laboratory of Systems Biology of Pathogens, Institute of Pathogen Biology, and Center for AIDS Research, Chinese Academy of Medical Sciences & Peking Union Medical College, Beijing, P. R. China, 2 Department of Medical Oncology, Beijing Hospital, Beijing, P. R. China, 3 Institute of Medicinal Biotechnology, Chinese Academy of Medical Sciences & Peking Union Medical College, Beijing, P. R. China, 4 McGill Centre for Viral Diseases, Lady Davis Institute, Jewish General Hospital, Montreal, Canada

☯ These authors contributed equally to this work.
* chen.liang@mcgill.ca (CL); guofei@ipb.pumc.edu.cn (FG)

**Data Availability Statement:** All relevant data are within the manuscript and its Supporting Information files.

**Funding:** This study was supported by funds from CAMS Innovation Fund for Medical Sciences

## Abstract

Long interspersed element type 1 (LINE-1, also L1 for short) is the only autonomously transposable element in the human genome. Its insertion into a new genomic site may disrupt the function of genes, potentially causing genetic diseases. Cells have thus evolved a battery of mechanisms to tightly control LINE-1 activity. Here, we report that a cellular antiviral protein, myxovirus resistance protein B (MxB), restricts the mobilization of LINE-1. This function of MxB requires the nuclear localization signal located at its N-terminus, its GTPase activity and its ability to form oligomers. We further found that MxB associates with LINE-1 protein ORF1p and promotes sequestration of ORF1p to G3BP1-containing cytoplasmic granules. Since knockdown of stress granule marker proteins G3BP1 or TIA1 abolishes MxB inhibition of LINE-1, we conclude that MxB engages stress granule components to effectively sequester LINE-1 proteins within the cytoplasmic granules, thus hindering LINE-1 from accessing the nucleus to complete retrotransposition. Thus, MxB protein provides one mechanism for cells to control the mobility of retroelements.

## Author summary

Retrotransposons occupy more than 40% of human genome, and have co-evolved with humans for millions of years. Long interspersed element type 1 (LINE-1, or L1) is the only retrotransposon that is able to jump to a new locus. LINE-1 retrotransposition causes genome instability, and is associated with genetic diseases including autoimmune diseases and cancer. To suppress this genome toxicity caused by LINE-1, humans have developed multi-layered mechanisms to control LINE-1 activity. MxB has been previously shown to inhibit LINE-1 mobility, thus contributing to host restriction of LINE-1. Here, we further demonstrate that MxB effectively restricts LINE-1 retrotransposition by sequestering LINE-1 ribonucleoprotein (RNP) within the cytoplasmic stress granules, thus guards

(CIFMS 2021-1-I2M-038) to F.G., from the National Key Plan for Scientific Research and Development of China (2018YFE0107600 and 2020YFA0707600) to F. G., from the Ministry of Science and Technology of China (2018ZX10301408-003) to F.G., from the National Natural Science Foundation of China (82072288) to F. G., from the Canadian Institutes of Health Research (CCI-132561) to C.L. and from the CAMS general fund (2019-RC-HL-012) to F. G.. The funders had no role in study design, data collection and analysis, decision to publish, or preparation of the manuscript.

**Competing interests:** The authors have declared that no competing interests exist.

genome stability. Hence our data attribute the restriction function of MxB to sequestering LINE-1 RNP to stress granules.

## Introduction

Non-long terminal repeat (LTR) retrotransposons include long interspersed element 1 (LINE-1), Alu and SINE-VNTR-Alu (SVA). They have proliferated over the past 80 million years in primates, and together account for approximately 45% of the human genome with LINE-1 alone taking 17% [1–3]. Among ~500,000 LINE-1 copies in human genome, about 80 to 100 LINE-1 elements are still capable of retrotransposition [1]. LINE-1 encodes two proteins called ORF1p and ORF2p. ORF1p is an RNA-binding protein and associates with LINE-1 RNA [4–7]. ORF2p has endonuclease and reverse transcriptase activities [8,9]. ORF1p, ORF2p and LINE-1 RNA together form an RNP complex that enters the nucleus where LINE-1 RNA is reverse transcribed and integrated into cellular DNA [10,11]. LINE-1 reverse transcription and integration is one continuous process, named target primed reverse transcription (TPRT) [12,13]. LINE-1 proteins also assist SVA and Alu mobilization [14,15]. Inevitably, retrotransposition creates various deleterious effects on the structure and function of the human genome [16]. Not surprisingly, more than 100 single-gene genetic diseases have been reported due to LINE-1 and Alu insertion [17,18].

Given the mutagenic nature of LINE-1 retrotransposition, cells have developed various defense mechanisms to restrict LINE-1 mobilization [19]. One arsenal of mechanisms are provided by innate immune factors that have been shown to inhibit virus infections. These include APOBEC3 (apolipoprotein B mRNA editing enzyme catalytic polypeptide 3) [20,21], TREX1 (Three Prime Repair Exonuclease 1) [22], MOV10 (Moloney leukemia virus type 10 protein) [23,24], SAMHD1 (SAM domain and HD domain containing protein 1) [25,26], RNase L (Ribonuclease L) [27], and ZAP (zinc-finger antiviral protein) [28,29]. These host factors operate by targeting LINE-1 RNP, and either directly degrading LINE-1 RNA by RNase L, mutating newly synthesized LINE-1 DNA by APOBEC3 proteins, degrading ORF1p by TREX-1, or sequestering LINE-1 RNP in the cytoplasmic granules by MOV10, SAMHD1 or ZAP. In 2015, Goodier and colleagues screened a panel of ISGs and restriction factors with known antiviral activities for effects of their expression on L1 retrotransposition. They demonstrated that BST2, ISG20, MAVS, MX2, and ZAP showed strong L1 inhibition [28]. However, the mechanism of MxB inhibiting LINE-1 has not been completely elucidated.

Mx proteins are dynamin-like large GTPases. They have been reported to inhibit distinct groups of viruses by different mechanisms. MxA is cytoplasmic protein, inhibits a plethora of viruses from diverse families, including influenza A virus, vesicular stomatitis virus, LaCrosse virus, hepatitis B virus and others [30]. The antiviral activity of MxA relies on the GTP binding and hydrolysis domain, oligomerization of the stalk domain and the intact BSE (bundle signaling element). In contrast, much fewer viruses have been reported to be restricted by MxB. These viruses include HIV-1 [31–33], HBV [34], HCV [35] and herpesviruses [36,37]. MxB is mainly observed at the nuclear envelope, inhibits HIV-1 and herpesviruses through targeting viral capsid and blocking nuclear import of viral DNA [38]. The N-terminal domain (NTD) and the stalk domain are crucial for MxB to intercept HIV-1. Our data demonstrate that MxB restricts LINE-1 retrotransposition by engaging the stress granule pathway.

## Results

### MxB inhibits LINE-1 retrotransposition

We used the CMV-L1-neo$^{RT}$ reporter to measure LINE-1 retrotransposition. This construct has a neomycin resistance gene inserted into the 3' untranslated region of LINE-1 in such a way that this gene can only be expressed from the reverse transcribed LINE-1 DNA, leading to cell resistance to G418 [39,40]. To test whether Mx proteins affect LINE-1 retrotransposition, we co-transfected HeLa cells with human Mx plasmid DNA and the CMV-L1-Neo$^{RT}$ reporter DNA (S1A Fig), followed by G418 selection. The results of colony assays showed that the ectopically expressed MxB diminished LINE-1 retrotransposition by 4.7-fold (p<0.01), whereas MxA exerted no effect (S1B Fig). We generated a defective L1 control which has the ORF1 RR261/262AA mutations inserted in the CMV-L1-neo$^{RT}$ DNA [40]. As a control for the specificity of MxB inhibition, the pEGFP-N1 vector DNA was tested with MxB overexpression, and no effect on the formation of G418 resistance cell colonies was observed (S1B Fig). A dose-dependent inhibition of LINE-1 by MxB was observed, with the lowest amount (200 ng) of MxB DNA causing significant loss of LINE-1 activity (S1C Fig). Cell viability was not affected by MxB overexpression (S1D Fig). Next, we performed quantitative PCR to measure levels of reverse transcribed LINE-1 DNA in cells transfected with CMV-L1-neo$^{RT}$ [41]. In agreement with the data of colony assay, MxB reduced LINE-1 DNA level by more than 2-fold (S1E Fig).

Although MxB is an interferon (IFN) stimulated gene, MxB has been shown to express at measurable levels in several human tissues such as lymph node, liver, and less expressed in brain and reproductive system (S2A Fig). When we used RT-qPCR and Western blot to measure endogenous MxB RNA and protein levels in HeLa cells, MxB expression in the absence of IFN treatment was detected as previously reported (S2C and S2D Fig) [42,43]. We then generated endogenous MxB knockout HeLa cell line using CRISPR/Cas9, and confirmed MxB knockout by sequencing (S2B Fig) and western blot (Fig 1D). We observed that LINE-1 activity increased by 1.9-fold (p<0.01) in the MxB knockout HeLa cells (Fig 1A). In addition, we measured the viability of MxB knockout cells and found no significant difference between MxB knockout and control cell lines (S2E Fig). The same observations were made with cells stably expressing MxB (S2F Fig). We further tested CMV-L1-neo$^{RT}$ transposition in HeLa cells stably expressing MxB, and observed 2.9-fold (p<0.01) decrease compared to that in the control cells (Fig 1B). Together, these data demonstrate an inhibition of LINE-1 restrotransposition by MxB, in agreement with previous report [28].

IFN has been shown to inhibit LINE-1 replication [44]. Since MxB is an IFN stimulated gene, we investigated the role of MxB in IFN-mediated LINE-1 inhibition. Activity of CMV-L1-neo$^{RT}$ in the control and MxB knockout cells was measured with and without IFN. The results of colony assay showed that LINE-1 retrotransposition was strongly inhibited by IFN in the control cell line, but this inhibition was significantly lost in MxB knock-out cell line (S2G Fig), which indicates that IFN-induced MxB contributes to IFN inhibition of LINE-1.

In support of MxB inhibition of LINE-1 retrotranspotion, the expression level of LINE-1 ORF1p diminished in HeLa cells stably expressing ectopic MxB (Fig 1C), and increased when MxB was knocked out (Fig 1D). We also investigated the effect of MxB on the activity of endogenous LINE-1 by performing quantitative RT-PCR to measure the level of endogenous LINE-1 RNA, using three pairs of primers that specifically amplify regions in the 5'UTR, ORF1 or ORF2 (Fig 1E) [45]. In MxB knockout HeLa cells, levels of endogenous LINE-1 RNA, as measured by all three primer pairs, significant increased, as opposed to the marked decrease seen in HeLa cells stably expressing MxB (Fig 1F and 1G). In summary, these results suggest that MxB is an inhibitor of LINE-1 retrotransposition.

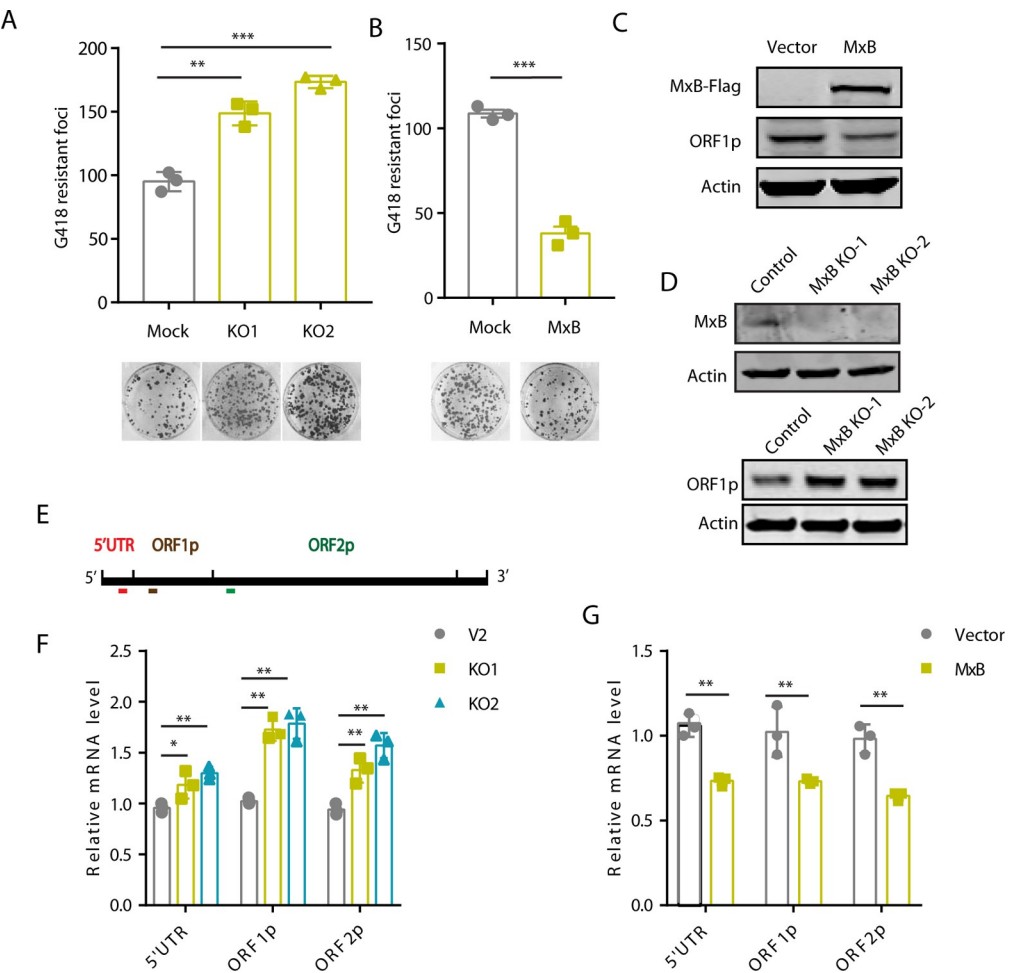

**Fig 1. MxB restricts L1 retrotransposition.** (A) MxB knockout (KO) HeLa cells were transfected with 250 ng CMV-L1-neo$^{RT}$ DNA. Neomycin-resistant cell colonies were scored, and the results of three independent experiments are presented in the bar graph (mean ± SEM; paired $t$-test) (B) MxB stably expressing HeLa cells were transfected with CMV-L1-neo$^{RT}$ DNA (250 ng). G418-resistant cell colonies were scored, and the results of three independent experiments are presented in the bar graph (mean ± SEM; paired $t$-test). (C) Western blots to measure the expression of ORF1p and MxB in HeLa cells stably expressing ectopic MxB. (D) Levels of ORF1p in MxB knockout (KO) HeLa cells. (E) Location of the amplified regions in the LINE-1 genome. Red, 5'UTR; brown, ORF1p; green, ORF2p. (F, G) Levels of LINE-1 RNA in HeLa cells stably expressing MxB or with MxB knockout, as determined by real-time RT-PCR. Data are the average from three independent experiments (mean ± SEM; paired $t$-test). *, P<0.05; **, P< 0.01; ***, P<0.001.

## MxB promotes sequestration of LINE-1 ORF1p to cytoplasmic bodies

We next investigated the molecular mechanisms by which MxB inhibits LINE-1 activity. In HeLa cells stably expressing MxB-EGFP, when the CMV-L1-neo$^{RT}$ reporter DNA was transfected, the ORF1p protein was seen to form cytoplasmic granules and co-localize with MxB (Fig 2A), which was also observed when only ORF1p protein was expressed in the absence of ORF2p (Fig 2B). These data were reproduced with transiently expressed MxB (S3A and S3B Fig).

We also investigated the relationship of subcellular location between MxB and LINE-1 RNP which were measured by the LINE-1-ms2x6/MS2-GFP system. As described previously, the expressed LINE-1 RNA contains 6 copies of MS2-binding sites which is bound by MS2-GFP

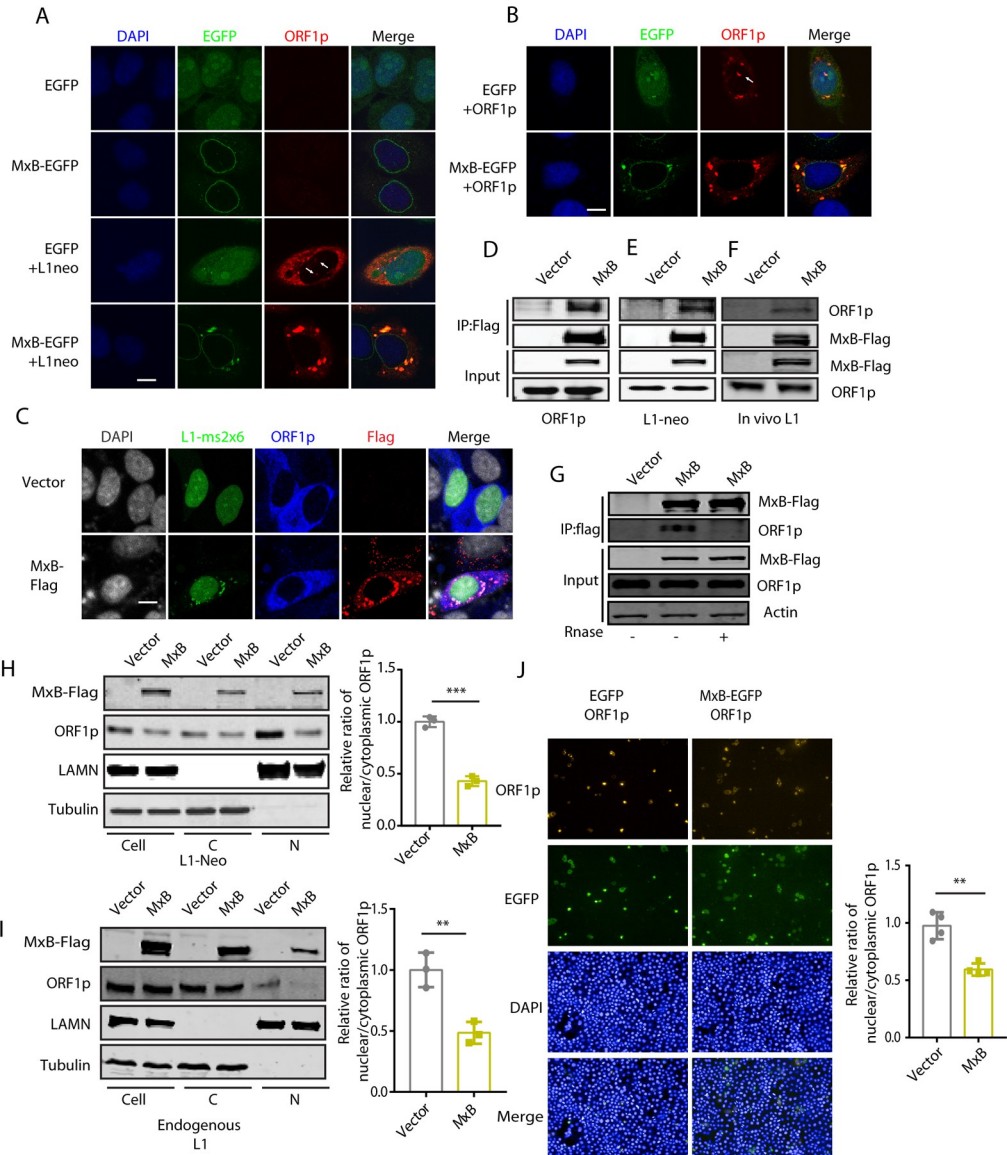

**Fig 2. MxB interacts with ORF1p and sequesters LINE-1 RNP within the cytoplasm.** (A) Immunofluorescence microscopy of MxB and ORF1p in HeLa cells which stably expressed MxB-EGFP and were transfected with CMV-L1-neo[RT] DNA. The white arrows indicate nuclear ORF1p. (B) Co-localization of stably expressed MxB-EGFP with ORF1p in HeLa cells that were transfected with ORF1p-Flag DNA. The white arrows show nuclear ORF1p. (C) Immunofluorescence microscopy of MxB and L1-ms2x6/MS2-GFP plasmids. The transcribed LINE-1 RNA contains six copies of MS2-binding site that are bound with MS2-GFP. The cytoplasmic punctate green fluorescence indicates localization of LINE-1 RNA. (D, E, F) Co-immunoprecipitation of transiently expressed MxB-Flag and ORF1p which was expressed either from the ORF1p-Myc DNA (D), the CMV-L1-neo[RT] DNA (E), or the endogenous LINE-1 in HEK293T cells (F). (G) Effect of RNase on co-immunoprecipitation of MxB and ORF1p. 293T cell were co-transfected with CMV-L1-neo[RT] vector and MxB-Flag. Co-IP was performed with or without RNase A. (H, I) Nucleocytoplasmic fractionation of HeLa cells transfected with the MxB DNA (48 hours). Presence of ORF1p, either ectopically expressed from the CMV-L1-neo[RT] DNA (H) or the endogenous form (I), in the nuclear (N) and cytoplasmic (C) fractions was detected by Western blots. In the Western blots, the nuclear fraction samples were from cells 5 times more than the cells of the cytoplasmic fraction samples. The intensities of protein bands were quantified and the results are presented in the bar graphs. To calculate the ratio of nuclear/cytoplasmic MxB, the amount of nuclear ORF1p normalized with the nuclear LAMN protein level, the amount of cytoplasmic ORF1p with tubulin, then nucleus/cytoplasm ORF1p ratio was calculated by dividing nuclear ORF1p value by the cytoplasm ORF1p value. At the end, the nucleus/cytoplasm ORF1p ratio of the vector control is arbitrarily set at "1". N, nucleus; C, cytoplasm (mean ± SEM; paired *t*-test) (J) The Operetta High-Content Screen system (PerkinElmer) was utilized to detect the ORF1p fluorescence signals within the nuclei and the cytoplasm in HeLa cells which were transfected with MxB-EGFP and ORF1p-Flag. Ratios of the nuclear

and cytoplasmic ORF1p signals were calculated. Results from the control group, which was transfected with the pEGFP-N1 vector, were arbitrarily set as 1. The results are summarized in the bar graph (mean ± SEM; paired *t*-test). **, P< 0.01; ***, P<0.001.

for visualization of LINE-1 RNA [46]. When LINE-1-ms2x6 and MS2-GFP DNA were co-expressed with MxB-Flag, LINE-1 RNA was seen to co-localize with MxB in the cytoplasm (Fig 2C). The association of MxB with LINE-1 RNP was also supported by the data of co-immunoprecipiation experiments performed with ectopic ORF1p (Fig 2D and 2E) or endogenous ORF1p (Fig 2F). The association of LINE-1 ORF1p and MxB was reduced when RNase was added during co-immunoprecipitation to remove RNA (Fig 2G). These results suggest that MxB associates with ORF1p in an RNA-dependent manner.

MxB has been shown to inhibit the nuclear import of HIV-1 DNA [43,47,48]. We asked whether MxB also impairs the nuclear import of LINE-1 RNP. To answer this question, we performed nucleocytoplasmic fractionation experiments to measure the levels of ORF1p in the cytoplasm and the nucleus. Indeed, MxB reduced the nuclear ORF1p which was either ectopically expressed or endogenous (Fig 2H and 2I). This observation was supported by the data of high-content imaging analysis (Fig 2J), which calculates the ratios of ORF1p signals in the nucleus and cytoplasm in HeLa cells. We could also observe nuclear ORF1p in control cells, which disappeared with MxB overexpression (Fig 2A and 2B, indicated by white arrows).

Once in the nucleus, LINE-1 ORF2p cleaves cellular DNA by its endonuclease activity to initiate reverse transcription. This DNA cleavage event causes DNA damage response, shown by the increase of γH2AX foci. If MxB prevents the nuclear import of LINE-1 RNP, we expected to detect fewer LINE-1-induced γH2AX foci with MxB expression. When we immunostained γH2AX, we detected a great number of γH2AX foci in cells transfected with the CMV-L1-neo^RT DNA. As expected, MxB stable expression dramatically reduced the number of these foci (Fig 3A, 3B and 3C). In agreement with the MxB overexpression data, greater numbers of γH2AX foci were scored in MxB knockout HeLa cells with CMV-L1-neo^RT transfection than in the control cells, concurrent with increased LINE-1 ORF1 expression (Fig 3D, 3E and 3F). When LINE-1 DNA was transfected into MxB knockout cell lines, γH2AX level and foci number increased compared with control cells (S4A, S4B and S4C Fig). Ectopic MxB decreased the γH2AX level induced by CMV-L1-neo^RT (Fig 3G). The defective L1-ORF1 (RR261/262AA) did not increase γH2AX level. We also observed that MxB overexpression alone diminished the number γH2AX foci, and MxB knockout itself led to an increase in γH2AX foci, which is likely a result of the suppression of endogenous LINE-1 by MxB. These data further suggest that MxB reduces the formation of double-strand DNA breaks associated with LINE-1 retrotransposition.

## Knockdown of stress granule marker proteins abrogates MxB inhibition of LINE-1

LINE-1 ORF1p has been reported to associate with stress granules [49–51]. Several LINE-1 inhibitors have also been reported to co-localize with LINE-1 ORF1p in stress granules, including SAMHD1 [26], ZAP [28], APOBEC [52,53] and MOV10 [51,53]. We therefore asked whether the MxB- and ORF1p-containing cytoplasmic foci also bear stress granule markers. To this end, we performed immunofluorescence staining experiment to detect stress granule marker G3BP1 or TIA1. Indeed, the MxB/ORF1p foci contained G3BP1 and TIA1 in HeLa cells that stably expressed MxB and were transfected with CMV-L1-neo^RT reporter or ORF1p plasmid (Figs 4A, 4B, S5A and S5B). We have scored the number of ORF1p-containing stress granules with and without MxB expression, observed no significant difference between the

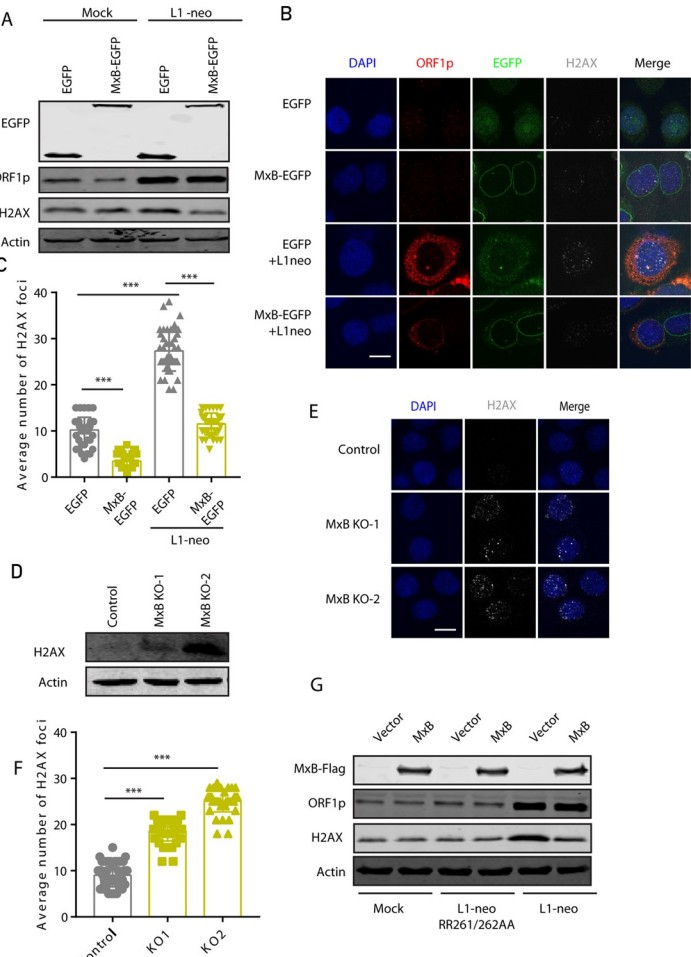

**Fig 3. MxB decreases the nuclear γH2AX foci induced by L1 retrotransposition.** (A) Western blots to show the levels of γH2AX in HeLa cells which stably express MxB and were transfected with the CMV-L1-neo$^{RT}$ DNA for 48 hours. (B) Detection of γH2AX foci in HeLa cells which stably express MxB and were transfected with the CMV-L1-neo$^{RT}$ DNA. Immunofluorescence was performed 24 hours post transfection. (C) The γH2AX foci were scored in more than 50 cells for each treatment. The average number of γH2AX foci per cell is presented in the bar graph (mean ± SEM; paired $t$-test). (D) Western blots to detect γH2AX in the control or MxB knockout HeLa cells without transfection of LINE-1 DNA. (E, F) Detection of γH2AX foci in the control and MxB knockout cells without LINE-1 DNA transfection. The γH2AX foci were scored in 50 cells, the results are presented in (F) (mean ± SEM; paired $t$-test). $^{**}$ indicates $P<0.01$; $^{***}$, $P<0.001$. (G) Western blot was performed to detect γH2AX in cells transfected with CMV-L1-neo$^{RT}$ or inactive L1-ORF1 (RR261/262AA), and MxB-Flag.

stable MxB-expressing cell line and control cell line (Figs 4A, 4B, S5A and S5B). However, we observed the enlarged ORF1p-containing foci under MxB-EGFP overexpression by visually inspecting more than 100 cells. The co-localization was also observed in HeLa cells that were transiently transfected with MxB plasmid DNA (S6A, S6B, S6D and S6E Fig). Moreover, the LINE1-ms2x6 RNA was also detected in the MxB/G3BP1 (Fig 4C) or TIA1 (S5C Fig) cytoplasm foci. In support of the localization of MxB, ORF1p with G3BP1 and TIA1, these four proteins were co-immunoprecipitated with each other, which was not observed with MxA (Fig 4D).

We next tested whether MxB inhibition of LINE-1 depends on stress granule formation. We thus depleted endogenous G3BP1 or TIA1, and examined whether MxB still inhibits LINE-1. Results of LINE-1 reporter assays showed that MxB lost inhibition of LINE-1 when

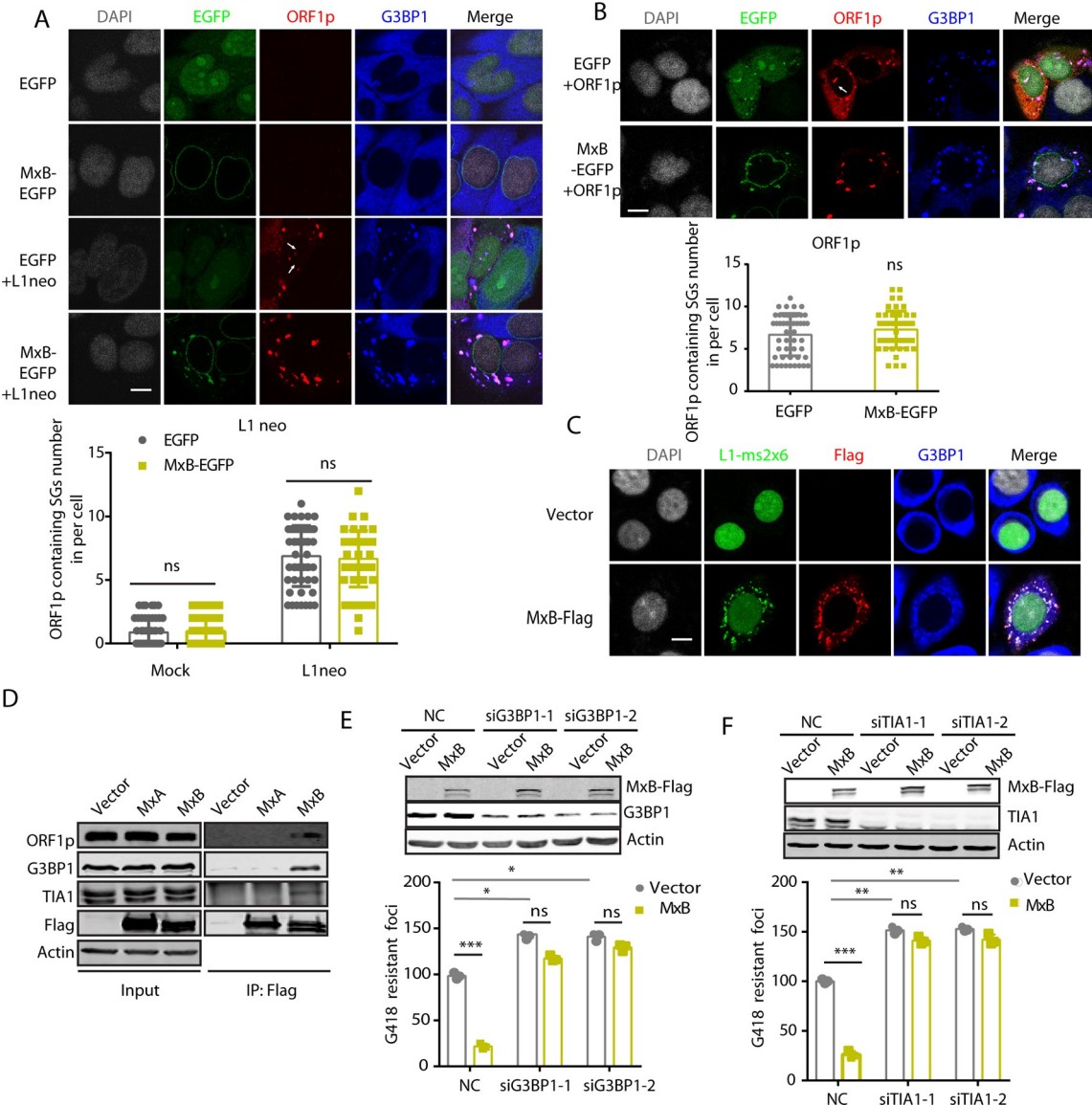

**Fig 4. Inhibition of LINE-1 by MxB depends on the stress granule pathway.** (A, B) Co-localization of stably expressed MxB-EGFP with ORF1p and G3BP1 in HeLa cells. ORF1p was either expressed from the transfected CMV-L1-neo^RT DNA (A) or ORF1p vector DNA (B). White arrows indicates nuclear ORF1p. ORF1p/G3BP1-containing SGs were scored in more than 50 cells for each treatment. The average number of ORF1p-containing SGs per cell is presented in the bar graph (mean ± SEM; paired *t*-test). (C) Co-localization of LINE-1 RNA, MxB-Flag and G3BP1 in cells transfected with MxB-Flag (500 ng), LINE-1-ms2x6 (750 ng) and MS2-GFP (250 ng) DNA. The LINE-1 RNA bears 6 copies of MS2-binding sites which are bound by MS2-GFP and detected as cytoplasmic puncta. (D) Co-immunoprecipitation to detect the association of MxB with ORF1p in HEK293T cells which were co-transfected with CMV-L1-neo^RT and MxA-Flag or MxB-Flag DNA. (E, F) Effects of G3BP1 (E) or TIA1 (F) knockdown on MxB inhibition of LINE-1 in HeLa cells which were co-transfected with MxB and CMV-L1-neo^RT DNA. The number of G418-resistant cell colonies was determined for each condition. The results are presented in the bar graphs (mean ± SEM; paired *t*-test). ns, not significant. ** denotes $P<0.01$; ***, $P<0.001$.

either G3BP1 or TIA1 was knocked down (Fig 4E and 4F). We also observed 50% increase in LINE-1 retrotransposition with knockdown of either G3BP1 or TIA1 in the absence of MxB expression (Fig 4E and 4F) [26]. In agreement with these functional data, knockdown of endogenous G3BP1 or TIA-1 led to the loss of ORF1p/MxB-EGFP granules in the cytoplasm (S6C and S6F Fig), further supporting the stress-granule nature of the ORF1p/MxB foci.

Arsenite (AS) is commonly used to induce canonical stress granules. Interestingly, MxB did not localize to stress granules that were induced with arsenite, neither the endogenous MxB induced by IFN-α in HeLa cell (S7A Fig) nor ectopically expressed MxB (S7B Fig), which suggests that MxB itself is not recruited to arsenite-induced stress granules, but rather is located to G3BP1- and TIA1-bearing foci through interaction with LINE-1 ORF1p. Together, these data demonstrate that MxB associates with LINE-1 RNP in the stress granules, and enhances cytoplasmic sequestration of LINE-1 RNP. We treated both the control and MxB knockout cell lines with AS. The number of SGs increased significantly with arsenite treatment, yet there was no difference between MxB knockout and control cell lines (S7C Fig). When the CMV-L1-neo$^{RT}$ plasmid was transfected, the number of ORF1p-containing SGs did not differ between MxB knockout and control cell lines with or without arsenite induction (S7D Fig). These results suggest that arsenite -induced ORF1p-containing SGs is not affected by endogenous MxB expression.

The inhibition of HIV-1 and HCV by MxB is dependent on cyclophilin A (CypA) [35,54]. To answer whether CypA affects the MxB inhibition of LINE-1, we used shRNA of CypA to knock down endogenous CypA in HeLa cells and performed colony assay of cells transfected with MxB and CMV-L1 neo$^{RT}$ reporter DNA. The results showed that LINE-1 retrotransposition was inhibited by MxB as much in CypA knockdown cells as in the control cells (S8A and S8B Fig), which indicates that MxB does not require CypA to restrict LINE-1 retrotransposition. Fewer G418-resistant cell colonies were scored with CypA-knockdown, which indicates a positive role CypA in LINE-1 retrotransposition.

## MxB requires its NLS, GTPase activity and oligomerization to inhibit LINE-1

MxB has a GTPase globular domain and a stalk domain which mediates MxB oligomerization [55–61]. In addition, the N-terminal sequence of MxB has a nuclear localization signal (NLS) [31,62]. To understand the contribution of the different MxB domain to LINE-1 inhibition, we generated a series of MxB mutants (Fig 5A). We first deleted the N-terminal 25 amino acids which contain the NLS, and generated mutant MxBdel25. We also replaced this 25-amino acid sequence with the NLS from the SV40 large T antigen, this mutant is named MxBdel25+NLS. Lastly, we replaced the first 42 amino acids of MxA with the first 25 amino acids of MxB, to generate the MxA+B25 chimeric protein (Fig 5A). The subcellular localization of these Mx proteins were examined with confocal microscopy (Fig 5B). MxA was detected in the cytoplasm, while MxB was localized at the nuclear envelope as previously reported [43,63]. When the NLS in N-terminal 25 amino acids of MxB was deleted, the MxBdel25 mutant changed subcellular location to the cytoplasm, the distribution was the same as the natural short isoform of MxB [64]. Results of LINE-1 reporter assays showed that MxBdel25 lost the ability of inhibiting LINE-1 (Fig 5C). MxBdel25+NLS inhibited LINE-1 as strongly as the wild type MxB, and the MxBdel25+NLS mutant was detected at the peri-nuclear region, a distribution pattern not exactly the same as MxB itself (Fig 5B and 5C). Interestingly, the MxA+B25 protein, which has its first 42 amino acids changed to the first 25 amino acids of MxB, was localized to the peri-nuclear region, and did not inhibit LINE-1 (Fig 5B and 5C). These data suggest that the NLS is essential but not sufficient for Mx proteins to inhibit LINE-1.

It was reported that MxB is localized at cytoplasmic face of nuclear core complex (NPC) and interact with NPC protein NUP214, as well as transport receptor transportin-1 (TNPO-1) [47,65]. We thus investigated whether NUP214 or TNPO-1 are required for LINE-1 inhibition by MxB. We observed that TNPO-1, but not NUP214, was localized to the MxB/ORF1p cytoplasm foci (S8C and S8D Fig). When we knocked down endogenous NUP214 or TNPO-1, we

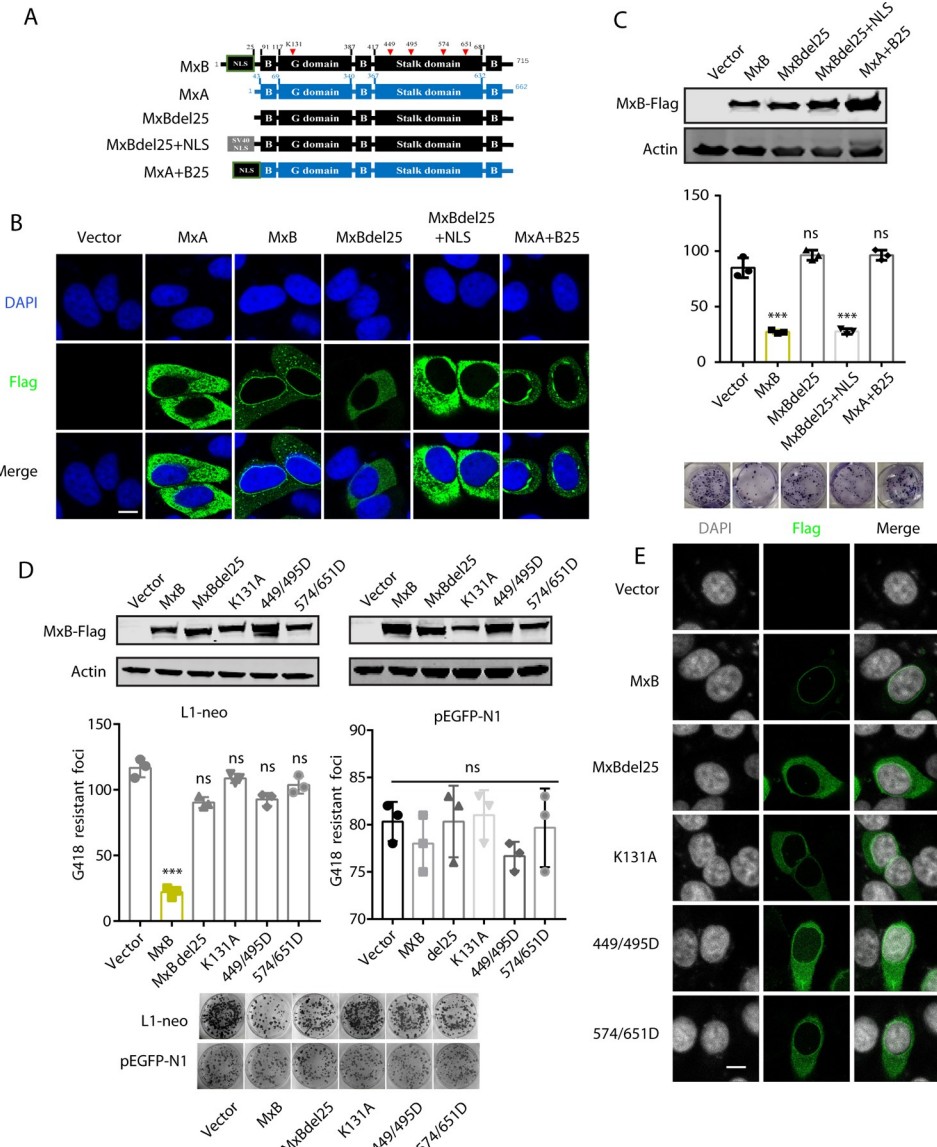

**Fig 5. Effect of MxB mutations on the inhibition of LINE-1 retrotransposition.** (A) Illustration of wild-type (WT) and mutated MxB proteins. NLS, nuclear localization signal; G domain, GTPase domain; B, bundle signaling element. (B) Subcellular localization of MxB and its mutants. (C) HeLa cells were co-transfected with CMV-L1-neo$^{RT}$ (250ng) and wild type MxB (250 ng) or its mutants, followed by G418 selection for resistant cell colonies. G418-resistant cell colonies were scored. The results of three independent experiments are presented in the bar graph (mean ± SEM; paired $t$-test). Expression of MxB and its mutants was determined by Western blot. (D) HeLa cells were co-transfected with CMV-L1-neo$^{RT}$ or pEGFP-N1 DNA and wild type MxB or the indicated MxB mutants. G418-resistant cell colonies were scored, the results of three independent experiments are presented in the bar graph (mean ± SEM; paired $t$-test). Expression of MxB and its mutants was determined by Western blot. **, $P < 0.01$; ***, $P < 0.001$. (E) MxB (500 ng DNA) or its mutants were transfected into HeLa cell. Their subcellular localization was detected by immunofluorescence staining. Bars represents 10 μm.

found that NUP214 knockdown led to moderate decrease of LINE-1 retrotransposition, whereas TNPO-1 knockdown moderately increased LINE-1 retrotransposition (S8E and S8F Fig). Knockdown of neither NUP214 nor TNPO-1 changed MxB inhibition of LINE-1. These data suggest a modest role of NUP214 and TNPO-1 in LINE-1 retrotransposition, but neither of these two proteins affect MxB inhibition of LINE-1.

We next generated more MxB mutants, including K131A which is defective in GTP binding ability [43], M574D/Y651D which disrupt MxB dimerization, as well as mutation R449D/F495D that prevent MxB oligomerization [59,60]. When these MxB mutants were tested in LINE-1 reporter assays, none of them affected LINE-1 activity (Fig 5D), in agreement with the decreased association of these MxB mutants with G3BP1 or ORF1p (S9A and S9B Fig), and none of these mutants affected the nuclear import of LINE-1 RNP (S9C and S9D Fig). These MxB mutants exhibited dispersed cytoplasmic localization (Fig 5E). Together, these results suggest that both the GTPase activity and oligomerization are essential for MxB to inhibit LINE-1.

## Inhibition of LINE-1 by Mx2-like proteins

We next tested whether MxB homologs of different species can also inhibit LINE-1. We first established a phylogenetic tree of Mx proteins from mammals (S10 Fig). The results showed that Mx proteins from most mammals segregate into two lineages, MxA/Mx1 and MxB/Mx2, with the exception of the mouse Mx proteins, mouse Mx1 and Mx2 are orthologous with human MxA. Human MxB ortholog in mouse was lost in evolution. We then tested the anti-LINE-1 activity of several Mx proteins from the MxA-like and MxB-like groups. The results showed that MxB protein from Macaca mulatta (Mac) and African green monkey (AGM) strongly inhibited LINE-1 (Fig 6A), whereas murine Mx1 and Mx2 did not affect LINE-1 activity (Fig 6B). The Equus caballus Mx2 also inhibited LINE-1 retrotransposition, the Ovis aries Mx2 exhibited week inhibition of LINE-1 (Fig 6B). To understand the variable effect of MxB-like proteins on LINE-1, we examined their subcellular localization. MacMx2 and AGMMx2 showed similar localization to the nuclear envelope as human MxB (Fig 6C). In the case of Equus and Ovis MxB proteins, their N-terminal sequences significantly differ from human MxB NLS. Nonetheless, Equus MxB was detected at the nuclear envelope as opposed to the Ovis MxB that was diffused in the cytoplasm (Fig 6C). These data suggest a correlation of localization to the nuclear envelope and the inhibition of LINE-1 by MxB-like proteins.

Finally, we tested whether human MxB can inhibit LINE-1 in non-human cells such as in murine BALB/3T3 clone A31 cell line. We stably expressed human MxB in the BALB/3T3 clone A31 cell line, and then transfected with human LINE-1 reporter CMV-L1-neo$^{RT}$ DNA. A strong inhibition of human LINE-1 retrotransposition was observed (Fig 6D). Levels of endogenous mouse LINE-1 subfamilies (L1A, L1Gf and L1Tf) and SINE RNA were measured by RT-qPCR with promoter specific primers in BALB/3T3 clone A31 cells that stably express human MxB. Marked inhibition of these mouse LINE-1 subfamilies (L1A, L1Gf and L1Tf) and SINE was observed (Fig 6E). IAP (Intracisternal A-type Particle elements) and MusD are active mouse ERVs (endogenous retroviruses) (also named LTR (long terminal repeat) retrotransposons) [66]. When we transfected IAP-neo$^{TNF}$ and MusD-neo$^{TNF}$ reporter DNA into human MxB-expressing BALB/3T3 clone A31 cells, the retrotranspostion of mouse IAP and MusD was strongly inhibited (Fig 6F). These results suggest that Mx2-like (orthologous to human MxB) proteins inhibit a wide range of retrotransposons in different species, thus play an important role in safeguarding the genome.

## Discussion

LINE-1 has approximately 100 full-length copies in human genome. Because of the mutagenesis nature of its retrotransposition into new loci, LINE-1 activity is tightly regulated by the host. Among the multi-layered mechanisms humans have evolved to control LINE-1 retrotransposition, it is not surprising that the innate antiviral factors, which defend against exogenous viruses, also have the ability of inhibiting LINE-1. This concept is strongly supported by a

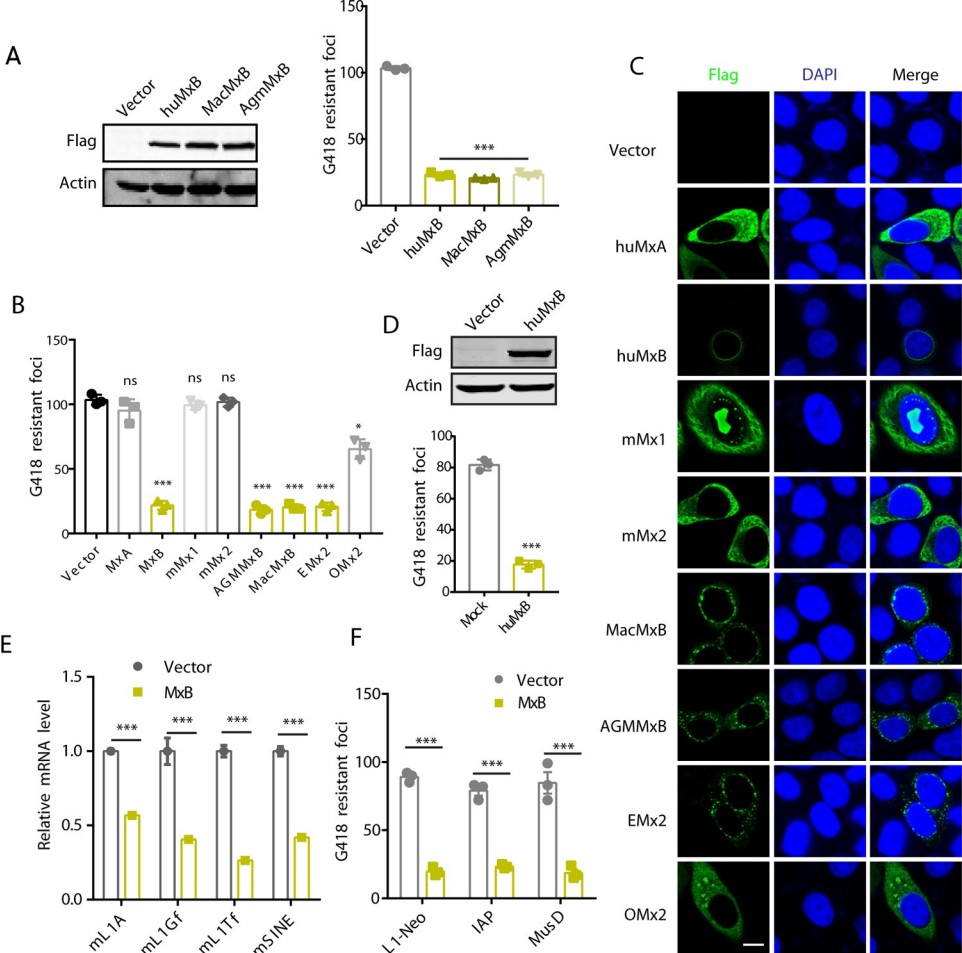

**Fig 6. Inhibition of LINE-1 by Mx2-like proteins.** (A) CMV-L1-neo[RT] reporter assay to measure the effect of non-human primate MxB on LINE-1 retrotransposition. Expression of MxB was determined by Western blot. (B) Effect of MxA-like and MxB-like proteins on LINE-1 activity. hu: human, m: mouse, Mac: Macaca mulatta, AGM: African green monkey, E: Equus caballus, O: Ovis aries. (C) Subcellular localization of MxB-like proteins in HeLa cells, detected with immunoflourescence microscopy. (D) Effect of stably expressed human MxB-Flag on CMV-L1-neo[RT] in the BALB/3T3 clone A31 cell line, as determined by colony assay. (E) Effect of stably expressed human MxB protein on mouse endogenous retroelements in the BALB/3T3 clone A31 cell line, as determined by RT-qPCR. Vector control was shown as grey dot, MxB data are shown as yellow square. (F) Effect of human MxB on mouse endogenous retroviruses reporter, IAP-neo[TNF] or MusD-neo[TNF], in BALB/3T3 clone A31 cells. G418 was added 48 hours post transfection. G418-resistant cell colonies were scored, and the results of three independent experiments are presented in the bar graph. Vector control was shown as gray dot, MxB data as yellow square. All data were plotted as mean values, with variation as SEM. Statistical significance was calculated by Student's two-tailed t test. *, P<0.05; **, P< 0.01; ***, P<0.001.

study from Goodier and colleagues who systematically tested a group of known interferon-induced antiviral proteins and found several of these factors with anti-LINE-1 function, including BST2, ISG20, MAVS, ZAP, and MxB (also called Mx2) [28]. In this study, we confirmed LINE-1 inhibition by MxB and further showed that MxB does so by sequestering LINE-1 RNP within cytoplasmic bodies containing stress granule marker proteins G3BP1 and TIA1. We further showed that the LINE-1 inhibition ability of MxB depends on its nuclear envelope subcellular location, the GTPase activity and oligomerization.

MxB has been shown to inhibit several important pathogenic viruses, including HIV-1 [31–33], herpesviruses [36,67], hepatitis B virus [34], hepatitis C virus, Japanese encephalitis virus, and

Dengue virus [35]. MxB exerts its antiviral function by targeting specific viral proteins. The capsids of HIV-1 [54,61,68–70] and herpesviruses [36,67] are the target of MxB. MxB inhibits HCV through binding to viral NS5A protein [35]. Our data showed that MxB is able to associate with LINE-1 protein ORF1p, enhancing the sequestration of LINE-1 RNP in G3BP1-containing cytoplasmic granules. In analogy to MxB inhibition of the nuclear import of HIV-1 and herpesvirus DNA, MxB sequesters LINE-1 RNP within the cytoplasm, thus prevents LINE-1 RNP from accessing the nucleus to complete reverse transcription. Therefore, MxB manages to use similar strategies to suppress virus infection and LINE-1 retrotransposition.

Clearly, there are differences in the detailed mechanisms by which MxB inhibits different viruses and LINE-1. For example, the requirement for the GTPase activity differs. The GTPase defective MxB still inhibits HIV-1 [31,32,54] but not herpesviruses [36,67]. Our data showed that MxB needs its GTPase function to suppress LINE-1. Although MxB depends on the cellular peptidylprolyl isomerase, cyclophilin A (CypA), to inhibit HIV-1 [33,54] and HCV [35], MxB inhibition of LINE-1 is CypA-independent (S8A and S8B Fig). These different requirements may reflect the different natures of the viral proteins which MxB targets, as well as the different replication strategies that the inhibited viruses employ. Regardless of these mechanistic variations, MxB strongly depends on its oligomerization and N-terminal sequence to inhibit its target viruses and LINE-1. It has been reported that MxB N-terminal sequence participates in the recognition of HIV-1 capsid structure and this activity is subject to regulation by phosphorylation [71,72]. It is possible that MxB interaction with LINE-1 ORF1p also requires its N-terminal sequence. We also noted that the N-terminal sequence was also reported to be dispensable for MxB to inhibit LINE-1 [28]. This discrepancy with our observation may be a result of the different experimental systems used in different studies including cell lines and protein tags. More studies are warranted to resolve this discrepancy.

Our data showed that MxB inhibition of LINE-1 depends on stress granule marker proteins G3BP1 and TIA1 (Fig 4E and 4F). LINE-1 proteins ORF1p and ORF2p as well as LINE-1 RNA have been reported to associate with cytoplasmic granules containing stress granule markers and p body markers [49–51]. This suggests that sequestering RNA to stress granules and p bodies serves as an intrinsic cellular strategy to control the activity of LINE-1, which operates through sequestering LINE-1 RNP within cytoplasmic bodies and therefore preventing LINE-1 from accessing the nucleus and completing retrotransposition. However, this strategy by itself may not be quite effective, as knockdown of G3BP1 or TIA1 only marginally increased LINE-1 activity (Fig 4E and 4F). MxB appears to enhance this intrinsic anti-LINE-1 mechanism, since we observed much greater LINE-1 ORF1p localization to G3BP1 or TIA1/MxB-positive cytoplasmic granules compared to the relatively more dispersed cytoplasmic distribution of ORF1p in the absence of MxB expression (Figs 4A, 4B, S5A and S5B). MxB is not the only factor stimulating sequestration of LINE-1 RNP in cytoplasmic granules, we and others have reported that ZAP [29] and SAMHD1 [26] employ a similar mechanism to assist cellular control of LINE-1 retrotransposition. It appears that cells have evolved means to regulate the stress granule-based anti-LINE-1 mechanism. One possible scenario is that high-level LINE-1 activity may stimulate the expression of MxB and other anti-LINE-1 factors that in turn suppress LINE-1 retrotransposition through engaging the stress granule pathway. In retrospective, sequestration of MxB to stress granules as a result of interaction with ORF1p alters subcellular localization of MxB, and may potentially influence its other cellular functions.

Stress granules are a typical membraneless organelles (MLOs) which are formed by liquid-liquid phase-separated in cytoplasm. MLOs regulate diverse cellular function, such as protein turnover, mitosis mRNA storage and translation, virus replication, and antiviral activity. Murine Mx1 and Mx2 were observed in nuclear bodies and granular cytoplasmic structures [73]. Coincidentally, exogenously expressed as well as IFN induced human MxA protein also

forms as membrane-less condensates in the cytoplasm, and interacts with and sequesters viral nucleocapsid proteins. Endogenous human MxB is localized to the cytoplasmic face of nuclear pore which also has phase-separated nature. Therefore, we speculate that Mx family proteins all have the propensity of forming MLO, which contributes to their antiviral function. MxB is not an integral component of stress granules, since we did not observe MxB localization with stress granules that were induced with arsenite treatment (S7A and S7B Fig). Expression of MxB itself did not cause stress granule formation either. Although there are MxB-containing cytoplasmic foci with MxB overexpression as seen in S3A Fig, these likely result from MxB aggregation. We suspect that MxB association with ORF1p stimulates G3BP1/TIA1-dependent aggregation of ORF1p, either through the GTPase activity of MxB and/or the ability of MxB to oligomerize. This mechanism allows MxB to inhibit LINE-1, also sends MxB to the ORF1p/G3BP1/TIA1 positive cytoplasmic granules.

Our data showed that the GTPase mutation K131A prevents MxB localization to the nuclear envelope and impairs its interaction with ORF1p. Furthermore, Betancor and colleagues reported that the GTPase domain cooperates with the N-terminal domain in Mx2 binding to the HIV-1 capsid [61]. We speculate that the GTPase activity may influence MxB conformation which is involved in interaction with LINE-1 ORF1p (S9C Fig) and inhibiting LINE-1 retrotransposition (Fig 5D).

We further investigated Mx proteins from different species for their ability to inhibit LINE-1. Mx2 proteins from non-human primates, including Macaca mulatta (Mac) and African green monkey (AGM), which are homologous to human MxB, also effectively restrict LINE-1 (Fig 6A), whereas Mx1 and Mx2 proteins from rodents, both of which share homology with human MxA, exhibit no effect (Fig 6B). Equus caballus Mx2 strongly inhibited LINE-1, whereas the Ovis aries Mx2 exhibited modest inhibitory effect (Figs 6B and S10). We also observed that anti-LINE-1 Mx2 or MxB proteins are located to nuclear envelope. Furthermore, classical nuclear localization signal of the SV40 large T antigen enables the MxBdel25 mutant to inhibit LINE-1 (Fig 5C). These results suggest a correlation of MxB-like protein localization to nuclear envelope and their anti-LINE-1 function.

Furthermore, the retrotranspostion of mouse SINE, IAP and MusD was strongly inhibited by stably expressed human MxB in BALB/3T3 clone A31 cells. This suggests that MxB regulates a wide spectrum of retroelements. SINE does not encode proteins, is unable to retrotransposition autonomously, its activity depends on LINE-1 ORF1p and ORF2p [14,15]. MxB is thus expected to inhibit SINE through sequestering LINE-1 ORF1p as we observed in this study. IAP and MusD are endogenous retroviruses, they encode Gag protein as HIV-1 does [74]. It is likely MxB targets Gag and inhibits IAP and MusD, similar to MxB inhibiting HIV-1 by targeting viral capsid. Since MxB also inhibits retrotransposons in murine cells, it can be envisioned that MxB either exerts its inhibition by itself or is assisted by factors shared by human and murine cells

In conclusion, our data support MxB as an anti-LINE-1 factor which functions by sequestering LINE-1 RNPs within the cytoplasmic granules through engaging the stress granule marker proteins G3BP1 and TIA1. This finding further substantiates the importance of the stress granule pathway in controlling LINE-1 and potentially other transposable elements. It will be interesting to investigate whether MxB uses similar mechanisms to restrict the infection of particular viruses.

## Materials and methods

### Plasmids and antibodies

CMV-L1-neo$^{RT}$ reporter DNA contains the complete human LINE-1 DNA and a neomycin resistance gene as a reporter of LINE-1 retrotransposition [26]. The ORF1-Myc DNA was

cloned into the pCMV-Tag 3B expression vector. The L1-ORF1 (RR261/262AA) plasmid was generated by inserting ORF1 261/262RR to AA mutations into the CMV-L1-neo$^{RT}$ plasmid, as reported in [40]. IAP-neo$^{TNF}$ and MusD-neo$^{TNF}$ reporter plasmids contain mouse LTR retrotransposon and a neomycin resistance cassette in reverse orientation [23]. The L1-ms2x6 contains six tandem MS2 CP binding sites near the 3' end of L1-RP in 99-PUR L1-RP vector. MS2-GFP plasmid was generated by inserting MS2 sequence in pEGFP-N1 vector. The MxB-Flag DNA was inserted into the pQC-XIP vector. The MxB mutations del25, K131A, R449D/F495D and M574D/Y651D were created using PCR-based mutagenesis method. The wild type and mutated MxB-EGFP were generated by fusing EGFP sequence to C-terminal end of MxB. The MxB sequence was inserted to the N-terminus of EGFP in pEGFP-N1 vector between the cleavage sites of restriction enzymes BamH1 and Age1, to generate MxB-EGFP fusion protein. Mouse Mx1 and Mx2, Equus caballus Mx2, Ovis aries Mx2, Mac and AGM Mx2 sequences were synthesized by Beijing Ruibio BioTech Co. Ltd company. These Mx cDNA sequences were cloned into the pQC-XIP vector.

Mouse anti-Actin antibody (66009-1-Ig), mouse anti-Myc antibody (67447-1-Ig), rabbit anti-Tubulin antibody (10094-1-AP), rabbit anti-G3BP1 antibody (13057-2-AP), rabbit anti-TIA1 antibody (12133-2-AP), and rabbit anti-TNPO-1 antibody (20679-2-AP) were purchased from Proteintech. Mouse anti-Flag (F1365) antibody, rabbit anti-Myc (C3956) antibody, rabbit anti-GFP antibody (G1544) and rabbit anti-LAMN1 antibody (L1293) were purchased from Sigma. Mouse anti-G3BP1 antibody (05–1938), mouse anti-ORF1p antibody clone 4H1 (MABC1152), mouse anti-γH2AX antibody (05–636) were purchased from MILLIPORE. Alexa fluor 555-labled donkey anti-rabbit antibody (A-21428) and Alexa fluor 647-labled donkey anti-mouse antibody (A-21236) were purchased from Thermo Fisher Scientific. The anti-MxB antibody was generated by immunizing rabbits with recombinant MxB protein [33]. Rabbit anti-ORF1p antibody was generated by immunizing rabbits with recombinant ORF1p [26]. Anti-Flag M2 affinity gel (A2220) was purchased from Sigma.

## Cell lines and cell culture

Human embryonic kidney cell line HEK293T expressing the SV40 T-antigen (ATCC, CRL-3216), human cervical carcinoma cell line HeLa (ATCC, CCL-2), and murine BALB/3T3 clone A31 (ATCC, CCL-163) cell line were grown in Dulbecco's modified Eagle's medium (DMEM), supplemented with 10% fetal bovine serum (FBS), 100 U/ml penicillin, 100 μg/ml streptomycin.

MxB KO cells were generated using the CRISPR-Cas9 system. Cells were transfected with lentiCRISPRv2 (52961, Addgene) [75] carrying single guide RNAs (sgRNAs) that target MxB. The gRNA sequence was listed in S1 Table. Following selection with puromycin (0.8 μg/ml), the resistant cells were serially diluted in 96-well plates to obtain single cell clones.

HeLa and BALB/3T3 clone A31 cells stably expressing MxB were generated using the retroviral vector system pQC-XIP (Clontech) which expresses MxB.

## Cell viability assay

Cell viability was measured using the cell-counting kit-8 (96992, Sigma) according to the protocol from the manufacturer. MxB/EGFP stably expressing cell lines or MxB knockout cell lines were cultured in 96-well plates with $2 \times 10^3$ cells per well. The MxB DNA was transfected into HeLa cells using PEI before cell viability was assessed. 10 μl cell-counting kit-8 was added into each well. After 3 hours, OD450 was measured with a microplate reader (Multiskan FC, ThermorFisher).

## Cell colony assay

The LINE-1 retrotransposition assay was performed as described previously [23,26]. In brief, $1.5 \times 10^5$ cells/well were seeded in 6-well plates. After 18 hours, cells were transfected with the CMV-L1-neo$^{RT}$ reporter DNA with or without MxB DNA, using PEI (sigma). 48 hours post-transfection, G418 (0.75 mg/ml) was added to select for resistant cells because of LINE-1 retro-transposition. Twelve days after selection, the G418-resistant colonies were fixing with 4% paraformaldehyde, stained with 0.5% crystal violet, and scored. IAP-neo$^{TNF}$ or MusD-neo$^{TNF}$ reporter assays were similarly performed as the CMV-L1-neo$^{RT}$ reporter assay.

## Quantification of LINE-1 cDNA by PCR

HeLa cells were transfected with MxB or empty vector and CMV-L1-neo$^{RT}$ reporter DNA.72 hours after transfection, total cellular DNA was extracted using the QIAamp DNA Mini kit (QIAGEN). The same amounts of DNA (250 ng) were subjected to PCR with primers L1cDNAF/L1cDNAR to amplify the reverse transcribed LINE-1 cDNA [46]. The forward primer L1cDNAR crosses neomycin resistance gene junction, thus only amplifies the spliced and reverse transcribed DNA. Levels of β-globin DNA were determined in PCR using primers β-globinF/ β-globinR. The results were used to normalize the levels of LINE-1 DNA [76]. The PCR products were separated in 1% agarose gels, and stained with Ethidium Bromide. All primer sequences are listed in S1 Table.

## Quantification of LINE-1 RNA by RT-qPCR

HeLa cells were transfected with CMV-L1-neo$^{RT}$ reporter DNA, with or without MxB DNA. 36 hours later, cellular total RNA was extracted using Trizol reagent (Invitrogen). The same amount of total cellular RNA was treated with DNase. The RNA was dissolved in RNase free water and reverse transcribed with SuperScript III Reverse Transcriptase (Invitrogen) according to the manufacturer's instruction with the oligo dT primer. Endogenous LINE-1 RNA levels were determined by qPCR using the Luna Universal qPCR Master Mix (NEB, M3003), and analyzed with the ΔΔCT method. The primers were named 5'UTRF/5'UTRR, ORF1pF/ ORF1pR, ORF2pF/ ORF2pR [45]. The GAPDH RNA was measured as the internal control. RNA levels of endogenous retrotransposition elements of mouse LINE-1 and SINE in BALB/ 3T3 clone A31 cells was measured by RT-qPCR with primers: mL1_TfF/mL1_TfR, mL1_GfF/ mL1_GfR, mL1_AF/mL1_AR, mSineF/mSineR. The internal control is mRrm2 [77]. All primer sequences are listed in S1 Table.

## Gene silencing

To knock down target gene by siRNA, HeLa cells was transfected with siRNA oligos (30 nM) by Lipofectamine RNAiMAX (Invitrogen) before plasmid transfection in 6-well plates. The knock-down efficiency was examined by Western blot. The siRNAs used were listed in S1 Table. The expression of CypA was silenced by shRNA targeting CypA (TRCN0000049228; Sigma).

## Immunofluorescence microscopy

Cells were grown on coverslips before transfection with the indicated plasmid DNA. 24 hours after transfection, cells were washed with phosphate buffered saline (PBS) (pH7.2), fixed with 4% paraformaldehyde for 10 min at room temperature, followed by a 10 min permeabilization with 0.3% TX-100 at room temperature. Cells were then blocked with 5% BSA (bovine serum albumin) in PBS, and further incubated for 2 hours with primary antibodies (anti-ORF1p (1:1000 dilution), anti-γH2AX (1:1000 dilution) or anti-G3BP1 (1:1000)) at room temperature.

Alexa fluor 555-conjugated donkey anti-rabbit antibody (A-21428) and Alexa fluor 647-conjugated donkey anti-mouse antibody (A-21236) were used as secondary antibodies. Confocal images were recorded with a Leica TCS SP5 (Leica Microsystems) mounted on an inverted microscope (DMI6000; Leica Microsystems), with an oil immersion 63x/NA1.4 objective len (HCX PL APO CS; Leica Microsystems).

## Immunoprecipitation

Transfected HEK293T Cells were harvested and lysed in the RIPA buffer (0.1% SDS, 1% Triton X-100, 1% sodium deoxycholate, 150 mM NaCl, 10 mM Tris [pH 7.5], 1 mM EDTA). The cell lysates were clarified by centrifugation at 12,000 rpm for 10 min at 4˚C. One milligram of the cell lysates was incubated with 50 µl of anti-Flag M2 affinity gel (A2220 Sigma) for overnight at 4˚C. The beads were then washed three times with the RIPA buffer, followed by incubation with the 2X loading buffer at 95˚C for 8 minutes to elute the bound proteins. The eluted materials were separated in SDS-12% PAGE, and further analyzed in Western blotting.

## Operetta high-content screening

To quantify the distribution of LINE-1 RNP complex in the nucleus and the cytoplasm, cells were visualized and analyzed on an Operetta High-Content Screen system (PerkinElmer). HeLa cells were transfected with ORF1-Flag and pMxB-EGFP. 48 hours after transfection, cells were seeded in a Cell- Carrier-96 plate (6005550, PerkinElmer) in triplicates at 20000 per well. 24 hours later, cells were fixed and permeablilized, and then incubated with DAPI to stain the nuclear DNA for 10 min at room temperature. After washing with PBS, plates were scanned using the Opertta HTS imaging system (PerkinElmer) to collect images which were further analyzed using the harmony software to quantify the percentage of ORF1p in the nucleus the cytoplasm in each cell [26].

## Cytoplasmic and nuclear fractionation

Subcellular fractionation was performed as described previously [78] Briefly, cells were harvested and lysed in buffer A (20 mM Tris, pH 7.6, 0.1 mM EDTA, 2 mM $MgCl_2$, 0.5 mM NaF, 0.5 mM $Na_3VO_4$ supplemented with protease inhibitors (Sigma-Aldrich, S8830)) for 2 min at room temperature and for another 10 min on ice. Nonidet P-40 (NP-40) was then added at a final concentration of 1% (vol/vol). The lysates were homogenized by gently vortexing or inverting the tube. Cytoplasmic fraction was collected by centrifugation in a pre-chilled centrifuge at 500xg for 3 min at 4˚C. The pelleted nuclei were washed three times in buffer A containing 1% NP-40, then suspended in one to two pellet volumes of the extraction buffer B (20 mM HEPES (pH 7.9), 400 mM NaCl, 25% (vol/vol) glycerol, 1 mM EDTA, 0.5 mM NaF, 0.5 mM $Na_3VO_4$ and 0.5 mM DTT), with vigorous vortex. The mixture was snap-frozen twice in liquid nitrogen and incubated for 20 min on ice. Lastly, the soluble nuclear extracts were collected by centrifugation at 20,000xg for 20 min at 4˚C. To visualize nuclear ORF1p, we examined in the Western blots nuclear fraction samples from cells 5 times more than the cytoplasmic fraction samples.

To calculate the ratio of nuclear/cytoplasmic MxB, we first normalized the amount of nuclear ORF1p with the nuclear LAMN protein level, the amount of cytoplasmic ORF1p with tubulin, then nucleus/cytoplasm ORF1p ratio was calculated by dividing nuclear ORF1p value by the cytoplasm ORF1p value. At the end, the nucleus/cytoplasm ORF1p ratio of the vector control is arbitrarily set at "1".

## Western blotting

Cells were lysed in the RIPA buffer containing 0.1% SDS, 1% Triton X-100, 1% sodium deoxycholate, 150 mM NaCl, 10 mM Tris (pH 7.5), and 1 mM EDTA. The lysates were separated in SDS-polyacrylamide gel (12%) (WB1103, Beijing Biotides Biotechnology Co. Ltd.). Proteins were transferred onto nitrocellulose membranes (Whatman). The membranes were probed with the indicated antibodies, followed by incubation with IRDye secondary antibodies (1:20000; LI-COR Biotechnology). Protein bands were visualized on a LI-COR Odyssey instrument (LI-COR Biotechnology). Intensities of protein bands were determined with ImageJ (National Institutes of Health) [79].

## Phylogenetic analysis

Orthologous Mx2 sequences were downloaded from Genbank. The protein sequences were aligned using Clustal Omega and converted to a codon alignment using MEGA 7. The Neighbor-Joining tree was calculated in MEGA 7 with 1,000 bootstrap replicates. The following Mx2 sequences were used: Homo sapiens MxB (NM_002463.1), Macaca mulatta (NM_001079696.1), Loxodonta Africana Mx2 (XM_023559110.1), Equus caballus Mx2 (XM_005606159.2), Sus scrofa Mx2 (NM_001097416.1), African green monkey Mx2 (XM_037984842.1), Myotis davidii Mx2 (XM_015569000.1), Bos taurus Mx2 (NM_173941.2), Ovis aries Mx2 (NM_001078652.1), Canis lupus familiaris Mx2 (XM_038443558.1), Vulpes vulpes Mx2 (XM_025983361.1), Mustela putorius furo Mx2 (XM_013061286.1), Ursus maritinus Mx2 (XM_008710207.2), Canis lupus familiaris Mx1 (NM_001003134.1), Eumetopias jubatus Mx1 (XM_028126469.1), Myotis davidii Mx1 (XM_006754326.2), Homo sapiens MxA (NM_001178046.3), Macaca mulatta Mx1 (NM_001079693.2), Sus scrofa Mx1 (NM_214061.2), Bos taurus Mx1 (NM_173940.2), Ovis aries Mx1 (NM_001009753.1), Mus musculus Mx1 (NM_010846.1), Rattus norvegicus Mx1 (NM_001271058.1), Mus musculus Mx2 (NM_013606.1); Rattus norvegicus Mx2 (NM_134350.2).

## Statistics

All experiments were performed three or more times independently under similar conditions. All data were plotted as mean values, with variation as SEM. Statistical significance was calculated by Student's two-tailed t test. P values of statistical significance are represented as *** $P<0.001$; ** $P<0.01$; * $P<0.05$.

## Supporting information

**S1 Fig. MxB diminishes L1 activity.** (A) Illustration of the L1-neo^RT reporter cassette. CMV-L1-neo^RT reporter DNA contains the complete human LINE-1 DNA and a neomycin resistance gene as a reporter of LINE-1 retrotransposition and a CMV promoter before 5'UTR. (B) HeLa cells were transfected with CMV-L1-neo^RT, defective L1-ORF1 (RR261/262AA) or pEGFP-N1 (carrying neomycin resistant gene) DNA together with MxA or MxB DNA. G418-resistant cell colonies were scored and results of three independent experiments are presented in the bar graph. Images of representative colony assays are shown. Ectopic expression of MxA-Flag or MxB-Flag was examined by Western blot. (C) HeLa cells were transfected with CMV-L1-neo^RT or pEGFP-N1 DNA together with increasing doses of MxB DNA. Results of three independent experiments are presented in the bar graphs. (D) Cell viability of HeLa cells transfected with different doses of MxB-Flag plasmid DNA for 48h. (E) HEK293T cells were co-transfected with the CMV-L1-neo^RT DNA and MxB DNA. Levels of the newly synthesized LINE-1 DNA were determined by semi-quantitative PCR. Levels of β-

globin DNA were measured as internal controls. Intensities of DNA bands were quantified, the results are summarized in the bar graph (mean ± SEM; paired *t*-test). *, $P<0.05$; **, $P<0.01$; ***, $P<0.001$.
(TIF)

**S2 Fig. MxB knockout with CRISPR-Cas9.** (A) MxB RNA levels in human tissues. Consensus Normalized eXpression (NX) levels for 55 tissue types and 6 blood cell types, created by combining the data from the three transcriptomics datasets (HPA, GTEx and FANTOM5), reference from www.proteinatlas.org. (B) The two guide RNA sequences are shown in red letters. The protospacer adjacent motifs (PAMs) are shown in orange letters. The mutated MxB sequences at the gRNA target sites are presented for each cell clone which was selected and used in this study. (C, D) Endogenous MxB protein and its mRNA level were determined with Western blot and RT-qPCR after stimulation with IFNα1 (25 ng/mL) for 16 hours. (E, F) Cell viability and cell growth rate of MxB knockout cell lines (E) and stable MxB-expressing cell line (F) were detected by cell-counting kit-8. (G) Colony assay was performed with MxB knockout or control cell lines which were transfected with CMV-L1-neo$^{RT}$ plasmid (500 ng) for 24 hours and treated with IFNα1 (2.5 ng/mL) for 24 hours. The results was presented in the bar graph (mean ± SEM; paired *t*-test). ns, not significant. *, $P<0.1$; **; $P<0.01$; ***, $P<0.001$.
(TIF)

**S3 Fig. Co-localization of MxB with L1 ORF1p.** (A, B) HeLa cells were co-transfected with MxB-EGFP and CMV-L1-neo$^{RT}$ reporter (A) or ORF1p-Flag DNA (B). Distribution of MxB and ORF1p was examined by immunofluorescence microscopy. The white arrows showed nuclear ORF1p. Fluorescence intensity analysis was performed at the position which indicated by white lines in merge panel to quantify the co-localization of EGFP or MxB-EGFP with ORF1p (A). Co-localization analyzed with fluorescence intensity used software from LAS AF (Leica). Scale bar, 10 μm.
(TIF)

**S4 Fig. CMV-L1-neo$^{RT}$ induces γH2AX in MxB knockout cells.** (A) Western blot to detect γH2AX in the control or MxB knockout HeLa cells which were transfected with CMV-L1-neo$^{RT}$ plasmid. (B, C) Detection of γH2AX foci in the control and MxB knockout cells transfected with CMV-L1-neo$^{RT}$ plasmid. The γH2AX foci were scored in 50 cells, the results are presented in (C) (mean ± SEM; paired *t*-test). ** indicates $P<0.01$; ***, $P<0.001$. Scale bar, 10 μm.
(TIF)

**S5 Fig. Co-localization of stably expressed MxB-EGFP with LINE-1 RNP and TIA1 in HeLa cells.** (A, B) Co-localization of stably expressed MxB-EGFP with ORF1p and TIA1 in HeLa cells. ORF1p was either expressed from the transfected CMV-L1-neo$^{RT}$ DNA (500ng) (A) or ORF1p vector DNA (500ng) (B) for 24 hours. White arrows indicate nuclear ORF1p. ORF1p/TIA1-containing SGs were scored in more than 50 cells for each treatment. The average number of ORF1p-containing SGs per cell is presented in the bar graph (mean ± SEM; paired *t*-test). ns, no significant. (C) HeLa cells were co-transfection with MxB-Flag (500 ng), LINE-1-ms2x6 (750 ng) and MS2-GFP (250 ng) plasmid DNA. Subcellular location of LINE-1 RNA was indicated by the binding of MS2-GFP to the 6 MS2-binding sites in the LINE-1 RNA. Scale bar, 10 μm.
(TIF)

**S6 Fig. Co-localization of MxB, ORF1p and G3BP1/TIA1 with transiently transfected MxB.** (A, B) Immunofluorescence microscopy to detect ORF1p, G3BP1 and MxB in HeLa

cells co-transfected with MxB-EGFP (500ng) and CMV-L1-neo$^{RT}$ DNA (500ng) (A) or the ORF1p-Flag DNA (500ng) (B) for 24 hours. The white arrows indicate nuclear ORF1p. (C) The MxB-EGFP stably expressing HeLa cell lines were transfected with siRNAs targeting G3BP1 and then transfected with CMV-L1-neo$^{RT}$ DNA (500ng). Immunofluorescence was performed to detect ORF1p, G3BP1 and MxB 24 hours after transfection. Expression of G3BP1 was examined by Western blot. Scale bar, 10 μm. (D, E) Immunofluorescence was performed to detect ORF1p, TIA1 and MxB in HeLa cells co-transfected with MxB and CMV-L1-neo$^{RT}$ DNA (500ng) (C) or the ORF1p-Flag DNA (500ng) (D). The white arrows indicate nuclear ORF1p. (F) MxB-EGFP stably expressing HeLa cells were transfected with siRNAs targeting TIA1 and then transfected with CMV-L1-neo$^{RT}$ DNA. Immunofluorescence was performed to detect ORF1p, TIA1 and MxB 24 hours post transfection. Expression of TIA1 was examined by Western blot. ORF1p-containing SGs were scored in more than 50 cells for each treatment. The average number of ORF1p-containing SGs per cell is presented in the bar graph (mean ± SEM; paired *t*-test) in (C and F). ***, $P<0.001$. Scale bar, 10 μm. (TIF)

**S7 Fig. MxB does not associate with stress granules induced by arsenite.** (A) HeLa cells were treated with IFN-α (25 ng/mL) for 24 hours, followed by exposure to arsenite (500 μM) for 30 min. G3BP1 and MxB were detected by immunostaining and fluorescence microscopy. (B) HeLa cells which stably express MxB-EGFP were treated with arsenite (500 μM) for 30 min. G3BP1 and MxB were detected by immunofluorescence microscopy. (C, D) MxB knockout cells were transfected CMV-L1-neo$^{RT}$ DNA (D) or vector control (C) for 24 hours, and then treated with arsenite (500 μM) for 30 min. SGs number was scored in more than 50 cells for each treatment, the results of three independent experiments are presented in the bar graph. The average number of SGs per cell is presented in the bar graph (mean ± SEM; paired *t*-test) in (C and D). ns, not significant. ***, $P<0.001$. Scale bar, 10 μm. (TIF)

**S8 Fig. MxB inhibits LINE-1 independent of NUP214, TNPO1 and CypA.** (A, B) CypA was knocked down before MxB and CMV-L1-neo$^{RT}$ were co-transfected into HeLa cells. Western blot was performed 48 hours post transfection to measure CypA and MxB levels (A). Meanwhile, G418 was added at 48 hours post transfection. G418 resistant colonies were scored and results of three independent experiments are presented in the bar graph (B). (C, D) Endogenous NUP214 (C), TNPO1 (D), CMV-L1-neo$^{RT}$ ORF1p and stably expressed MxB-EGPF were detected by immunofluorescence. (E, F) Effect of knocking down endogenous NUP214 (E) or TNPO1 (F) on MxB inhibition of CMV-L1-neo$^{RT}$ in HeLa cells. Results of three independent experiments are presented in the bar graph (mean ± SEM; paired *t*-test). Images of representative colony assays are shown. Expression of NUP214 and TNPO1 was examined by Western blot. ns, not significant. ** indicates $P<0.01$; ***, $P<0.001$. Scale bar, 10 μm. (TIF)

**S9 Fig. Effect of MxB mutations on association and subcellular localization with LINE-1 RNP.** (A) Immunofluorescence microscopy analysis of ORF1p, G3BP1 and MxB in HeLa cells co-transfected with CMV-L1-neo$^{RT}$ and MxB or its mutants. Scale bar, 10 μm. (B) 293T cells were co-transfected with CMV-L1-neo$^{RT}$ DNA and MxB-Flag or its mutants. Immunoprecipitation was performed with anti-Flag antibody 48 hours post transfection. Presence of ORF1p in the precipitated materials was detected by Western blot. (C) HeLa cells were co-transfected with CMV-L1-neo$^{RT}$ and MxB or its mutant DNA. Nuclear and cytoplasmic fractions were prepared and further examined in Western blot to determine the levels of ORF1p. The bar graph at the S9C right was the quantification of the immunoblot. (D) The Operetta High-

Content Screen system (PerkinElmer) was utilized to determine the ratios of the nuclear and cytoplasmic ORF1p. The results are summarized in the bar graph (mean ± SEM; paired *t*-test). ns, not significant. **, $P < 0.01$; ***, $P < 0.001$.
(TIF)

**S10 Fig. Phylogenetic analysis of of Mx genes from different species.** Neighbor-Joining method was used for the evolution analysis. Orthologous Mx2 sequences were acquired from Genbank. The protein sequences were aligned using Clustal Omega and converted to a codon alignment using MEGA 7. The Neighbor-Joining tree was calculated in MEGA 7 with 1,000 bootstrap replicates. Bootstrap values (>70) were also tested. Mx genes were selected from MxA-like and MxB-like major clades. "*" indicates the Mx genes that were investigated in this study for inhibiting LINE-1 retrotransposition.
(TIF)

**S1 Table. Primers and RNAs used in this study.**
(DOCX)

## Author Contributions

**Conceptualization:** Yu Huang, Fengwen Xu, Fei Guo.

**Data curation:** Yu Huang, Xiaoman Liu, Fei Zhao.

**Formal analysis:** Liang Wei, Zhangling Fan, Yamei Hu, Liming Wang.

**Funding acquisition:** Chen Liang, Fei Guo.

**Investigation:** Shan Mei, Bin Ai.

**Supervision:** Fengwen Xu, Shan Mei.

**Validation:** Yu Huang, Fengwen Xu, Xiaoman Liu.

**Writing – original draft:** Yu Huang, Fengwen Xu.

**Writing – review & editing:** Shan Cen, Chen Liang, Fei Guo.

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
