## [Decision Letter · Decision Letter 0]

10 Sep 2021

Dear Dr Guo,

Thank you very much for submitting your Research Article entitled 'MxB inhibits Long interspersed element 1 retrotransposition' to PLOS Genetics.

The manuscript was fully evaluated at the editorial level and by four independent peer reviewers. The reviewers appreciated the attention to an important problem, but raised some substantial concerns about the current manuscript. In particular, the reviewers found that the claims of relocalization or sequestration of ORF1p by MxB were not well supported by the immunofluorescence data as presented. Based on the reviews, we will not be able to accept this version of the manuscript, but we would be willing to review a much-revised version. We cannot, of course, promise publication at that time.

If you decide to revise the manuscript for further consideration at PLOS Genetics, please aim to resubmit within the next 60 days, unless it will take extra time to address the concerns of the reviewers, in which case we would appreciate an expected resubmission date by email to plosgenetics@plos.org.

[LINK]

We are sorry that we cannot be more positive about your manuscript at this stage. Please do not hesitate to contact us if you have any concerns or questions.

Yours sincerely,

Richard N McLaughlin, PhD

Guest Editor

PLOS Genetics

Gregory Barsh

Editor-in-Chief

PLOS Genetics

Reviewer's Responses to Questions

**Comments to the Authors:**

Reviewer #1: In this work Huang and colleagues investigate the inhibitory activity of the interferon stimulated protein MXB on the transposable element LINE-1. Authors describe how ectopically expressed MXB inhibits integration of LINE-1 in HeLa cells. Authors present evidence for the interaction and colocalization of MXB and ORF1P, in a complex that also colocalize with stress granule marker G3BP1. For inhibition to happen, MXB requires its N-terminal nuclear envelope localization signal, its GTPase and oligomerization activities, as well as the presence of stress granules components G3BP1 and TIA1. In a final set of experiments, they broaden the spectrum of MXB inhibition to other transposable elements such as IAP and Mus D.

This paper is interesting because it further widens the functions of MXB (MX2) as an "antiviral" factor, addressing its activity against retrotransposon elements. The notion that MXB itself has a potent inhibitory activity on restricting LINE-1 nuclear transport could be specially interesting for tissues with a “high” basal levels of MXB (such as granulocytes). In addition, it provides further insight on the tight regulation of transposable elements.

However, I think the paper has a series of weaknesses.

1. MXB is an Interferon (IFN) stimulated gene and in most cells, is virtually absent in the absent of IFN stimulation. On the other hand, LINE-1 has been linked to type 1 IFN induction before. However, the authors fail to make this connection. Is LINE-1 inducing an IFN response and then upregulating MXB? Or is this mechanism (LINE-1 inhibition) only valid for cells/tissues with some basal level of MXB? An easy experiment to link MXB to the inhibition of LINE-1 would be adding IFN to a CRISPR control cell line and to the MXB KO lines and quantify the extent of the IFN inhibition, where it should be reduced for the MXB KO lines.

2. Authors claim that nuclear envelope localization of MXB is an important determinant for the LINE-1 inhibitory mechanism. In the case of HIV-1, several articles have linked the importance of nuclear pore proteins to the MXB antiviral activity. I am missing some evidence on this regard. Are any nuclear pore proteins involved in LINE-1 inhibition by MXB? Are any nuclear pore proteins located to the MXB-containing stress granules?

3. Authors claim that MXB and ORF1P accumulates to stress granules and that transient depletion of G3BP1 or TIA1 depletes MXB of its inhibitory activity. I would like to see further confirmation of this. Specifically, will depletion of G3BP1 and/or TIA1 affect the accumulation of ORF1P and MXB to cytoplasmic bodies? I don’t think figures 4e and 4f really show the formation of stress granules.

4. I am not convinced by the microscopy throughout the paper, specifically:

a. Fig 2: MXB distribution should be the same in the EGFP panel for the MXB L1 NEO condition of fig 2a and the MXB-EGFP + L1 NEO panel from S3a, and in the Flag panel for the MXB-Flag condition in fig 2c. However, there is no MXB located to the nuclear envelope in fig 2c, while this is clearly noticeable in fig 2a and S3a.

b. Fig 6C: again, I am not convinced by this figure. None of the nuclear envelope-located proteins seem to be locating there. Staining with MXB and/or nucleoporins creates a tight ring around the nucleus (see for example Kane et al., 2018), while in here there is some space between the DAPI signal and the Flag signal, as if between the nucleus and the nuclear envelope was some “empty” space. In the case of human MXB there is not even such a ring, but just aggregation of protein in the proximity of the nucleus.

c. Lines 196-197: “in agreement with the inability of these MxB mutants to interact with G3BP1 or ORF1p, (Fig. S6A and B)”. This is not what I understand from S6a and S6b. Firstly, if I am not mistaken, in S6a colocalization of MXB, G3BP1 and L1 is indicated by the light-pink colour on the merge image. While this might be absent in the case of del25 and the M574D/Y651D mutant, I can see it in the K131A and R449D/F495D mutants.

5. Figure legends are too succinct. Some of the figures are virtually impossible to understand looking only at the legend, in others some graphics are not explained (as is the case of the arrows in microscopy images) and in most of them further explanation is needed to understand what they are showing.

In addition, I have other comments:

1. Author summary needs some correction, number of grammatical mistakes and apparent lack of text, such as:

a. Line 30: “Retrotransposons have been existed as ancient components”.

b. Line 32: “and inserts in genome by its endonuclease, which staggers”.

c. Line 35: “Among the risk of LINE-1 out of control, host develops”.

d. RNP is not defined

2. Line 70: HBV, a reference is missing.

3. Fig S1: It is not clear to me how the colony assay is quantified. It is not specified in M&M (where it probably should be), and in the legend of fig S1 it reads “Number of neomycin-resistant colonies with the control vector is arbitrarily set as 1”. I assume authors mean set as 100. Nevertheless, in figures depicting this assay (such as S1b and 1a &1b), the average of the vector control is not 100 (in S1b for example is clearly above it). Taking again S1b as an example, it seems that only one of the dots has a value of exactly 100, while the other 2 are clearly higher. Does this mean that one of the vector control experiments was set as a 100, and then all the other results (for vector and MX proteins) normalized to this value, rather than on each experiment setting the vector control to 100 and then normalizing the MX proteins to it?

4. Fig S1d doesn’t have a legend. In addition, -globin is misspelled as, -glubin.

5. Fig S2: The order of the different panels should be modified: S2e is presented in the text before any other panel from S2.

6. S2b: the level of MXB in both lines seems the same to me. One of the ref given on the text for previously showing presence of MXB in HeLa cells in the absence of IFN doesn’t show that, on the contrary, there is a total absence of MXB in the no IFN lane (see fig 1g from ref 36).

7. S2d: immunblot lacks enough quality to draw any conclusions from it.

8. Line 123: a “was” is missing.

9. Line 129: experiments instead of experiment.

10. Fig 2H: wouldn’t be expected an increased signal for L1ORF1 in the cytoplasm as the nuclear accumulation is reduced under MXB expression?

11. Fig 2I: I fail to see any signal for L1ORF1 in the nucleus.

12. Fig 2j: why is there less ORF1 p positive cells when expressing EGFP than when expressing MXB-EGFP?

13. Fig 3a: shouldn’t the level of L1ORF1 been reduced under MXB expression? This was shown in fig 1c.

14. Fig 3b: the intensity of the H2AX staining needs to be increased. It is quite difficult to see it as it is now.

15. Fig 3e: needs more intensity.

16. Fig 3g: I am not convinced by these results. First, what is mock here? No explanation in the figure legend. Second, why is there a marked increase of H2AX from mock to EN- if the level of ORF1P is the same? Third, I do see a small reduction of the H2AX signal in the EN- under MXB expression, is this statistically different from the reduction seen in the EN+ condition?

17. Fig 4a and 4b: the arrows shown are not mentioned in the figure legend, what are they signalling?

18. Fig 4b: why is the distribution of ORF1P more diffuse in the absence of MXB (also seem in S4a)? This is not the case in Fig 4a.

19. Fig 4c: Why is MXB not localize to the nuclear envelope here?

20. Lines 152-153: it reads “Indeed, the MxB/ORF1p foci had G3BP1 and TIA1 in HeLa cells that stably expressed MxB and were transfected with CMV-L1-neoRT reporter (Fig. 4A)”. However, TIA1 staining is not shown.

21. S4a: figure labelling contains a MXBE instead of MXB.

22. Fig 4d: it would enrich the paper seeing also the interaction with TIA1.

23. Line 185: MXB-B25 should be MXA-B25 or even MXA+B25, to keep consistency with line 181.

24. Fig 5a: there is no explanation for the MXB domains labelling.

25. Fig 5b: why is MXA aggregating in the cytoplasm? Previously it has been reported as a diffuse cytoplasmic protein (see for example Goujon et al., 2014 or Betancor et al., 2019) and it behaves this way in fig 6c. Also, none of the proteins with a nuclear envelope localization signal (MXB and MXA+B25) or an NLS (MXBdel25 + NLS) seems to accumulate at the nuclear envelope or at the nucleus.

26. In previous works, experiments using MXA chimaeras bearing the N-terminal domain of MXB and showing transfer of the antiviral activity have used MXA proteins where the whole N-terminal domain of MXA has been substituted with the whole N-terminal domain of MXB (see for example Goujon et al., 2014). I would recommend doing this experiment rather than attaching the first 25 amino acids of MXB to MXA, since this could potentially interfere with the function of the N-terminal domain.

27. Line 194: should say mutant M574D/Y651D instead of M574D, Y651D

28. Lines 194-195: since the authors are describing mutants, they should state they are using the R449D/F495D mutant, instead of saying they also mutate those 2 positions, since in the current format the text is not clear on whether they created 2 independent mutants or a double mutant (as they did).

29. Fig S6: I found confusing the arrangement of the figure. It seems like both bar charts at the bottom belong to S6d. The lack of any mention to the quantification of the immunoblots from S6c makes it more confusing. A further explanation on the figure legend is required.

30. The IP showed in S6b doesn’t show the lack of interaction of MXB mutants with L1ORF1: there is a clear signal for the latter in all the lanes and in the case of the vector control. Better experiments are needed to be able to make this claim.

31. Line 225: what are IAP and MusD?

32. Fig 6e: colour code description of the bars is missing in the figure.

33. Lines 257-258: “MxB inhibition of LINE-1 is CypA-independent (data not shown)”. While ideally such a claim should be accompanied by the data showing this, authors could further explained how they found this (CRISPR KO?, siRNA? Use of cyclosporine?, etc).

34. Line 288: remove “in” at the end of the line.

35. Line 289: a space is missing before “(65)”. Also see point 45.

36. Line 303: “Equus caballus Mx2, but not Ovis aries Mx2, inhibits LINE-1”. This claim is not supported by data in Fig 6B, since there are less colonies in cells expressing Ovis MX2 than in cells expressing MXA (with statistical significance).

37. Line 320: a “the” is missing in “D205A point mutation in endonuclease”.

38. Lines 321-322: the sentence “IAP-neoTNF, MusD-neoTNF reporter plasmids which contain mouse LTR retrotransposon and reverse direction neomycin resistance cassette(19)” is missing something.

39. In general, the “plasmids and antibodies” part of M&M needs rewriting since there are several grammatical mistakes.

40. Line 325: the mutation is M574 not N574.

41. Line 326: how is the MXB-EGFP generated? Is EGFP at the N or C-terminal end of MXB?

42. Line 333: is missing “antibody” at the end.

43. Line 335: is also missing antibody after “(05-1938)”.

44. Line 336: is Millipore

45. Throughout the text there are many spaces missing between the text and the reference call.

46. In “cell colony assay” an explanation on how the results are normalized is missing.

47. The way the number of hours is depicted is not consistent throughout the text. For example, in line 358 it says “18 hours”, while in line 367 it says “seventy-two hours”

48. Line 401: the “1,000” is the only time authors used a “,” all the other 1000 are without it. Also, some other numbers throughout the text are depicted with or without “,”. Consistency is needed.

49. Line 412: “were” should be “was”.

50. The M&M are in general too summarized. In most cases the amount of DNA transfected or the time after transfection is not indicated.

51. Line 432: “Na3VO4” should be written as Na3VO4.

52. Line 439: Should say “by centrifugation” instead of “with centrifugation”.

53. Line 443: What viruses?

Reviewer #2: In their manuscript “MxB inhibits Long Interspersed Element 1 retrotransposition”, Huang et al. demonstrate that the cellular antiviral protein MxB blocks L1 retrotransposition in cultured cells, and then investigate the mechanism by which this inhibition occurs using cultured cell retrotransposition assays, immunofluorescence imagining, and biochemistry approaches. Over all, the manuscript presents results that will be interesting to the L1 and virology fields. However, some claims are over-stated particularly with regards to the mechanism of inhibition, and some of the experiments could be presented more clearly particularly in the figures and legends.

Major points:

1. MxB was demonstrated to restrict L1 retrotransposition by Goodier and colleagues in 2015 (PMID 26001115). This publication is mentioned in the discussion of the present manuscript, but should be mentioned in the abstract or introduction. Otherwise the manuscript gives the initial impression that L1 inhibition by MxB is a novel finding. In addition, Doucet et al. (PMID: 20949108) used epitope-tagging strategies to investigate the subcellular localization of L1 proteins and RNA. This work should be cited.

2 While the results as a whole demonstrate that MxB mediates the localization of L1 ORF1p to stress granules, the conclusion that MxB inhibits L1 specifically by preventing L1 RNPs from entering the nucleus is not sufficiently supported. Figure 2 is titled with this claim, but the immunofluorescence images in A-C do not convincingly or quantitatively show ORF1p localizing to the nucleus in the absence of MxB, and previous studies have shown that wild-type ORF1p does not normally localize to the nucleus (PMID 29309036, PMID 20949108, PMID 20147320). Figure J claims to show this quantitatively using IF, but YFP (26.2 kDa) is a very bulky tag and is likely to affect cellular localization of ORF1p (40 kDa). This experiment would be much stronger if it utilized a smaller epitope tag already demonstrated to minimally impact L1 retrotransposition (eg, T7--PMID 20949108, PMID 16183655)

3 The fractionation experiment in panels H and I is a bit confusing: in H, the amount of ORF1p found in the nucleus with vector-only is dramatically more than in the cytoplasm, but this pattern is not supported by the immunofluorescence experiments in panel A. The quantitation shown in panels H and I should be better explained—the actual ratio of ORF1p in nucleus/cytoplasm for vector control, for example, is clearly not 1. How was this normalized? In addition, in panel I the detection of endogenous ORF1p in HeLa cells is not supported by previous studies, which describe HeLa cells to express very little or no ORF1p (PMID 20686575). Over all, the claim that MxB prevents nuclear import of L1 RNPs should be softened.

4 In the paragraph beginning with line 100, the authors say that the reduced detection of L1 RNA and ORF1p is consistent with MxB inhibiting retrotransposition. The rationale for this statement should be clarified—if the authors are proposing a sequestration model (rather than an inhibition of L1 expression), one would not necessarily expect the L1 RNA and protein levels to be affected.

5 Throughout the manuscript, the figure legends and labels, particularly on panels showing western blots, could be clearer and more informative. For example, many IF images have white arrows but it is never stated what these indicate. In figure 3A, the gel segment labeled “EGFP” shows detection in the MxB-EGFP transfection but not in the EGFP alone transfection. Presumably this is because the molecular weights of MxB-EGFP and EGFP alone are different, but this should be clarified. The figure legend for supplemental figure 1D is missing entirely.

6 Throughout the paper, sequestration of ORF1p by MxB is only shown by IF for exogenously expressed ORF1p. Given that the authors appear to have an antibody that apparently can detect endogenous ORF1p by western blot, it would greatly strengthen the paper to demonstrate that MxB sequesters endogenous ORF1p via immunofluorescence.

Minor comments

1. Line 54: cite a reference for the number of human genetic disease cases attributed to L1-mediated mutagenesis.

2. Line 69: Would be informative to discuss in more detail the mechanism(s) by which MxB and MxA inhibit viruses.

3. Line 86: state quantitatively what the “lowest amount” of plasmid was.

4. Figure 6D: which mouse L1 retrotransposition constructs were used? No papers cited and not shown in methods.

Reviewer #3: In this manuscript, Huang et al. invested the mechanism of MxB-mediated LINE-1 suppression. They started by confirming this reported phenomenon in a neomycin-resistance-based LINE-1 retrotransposition assay. Then IF and IP experiments were conducted, results of which suggested that MxB and LINE-1 ORF1p co-localizes in some dot-like organelles, which were later determined as stress granules. The authors further demonstrated the MxB suppresses LINE-1 replication through stress granule pathway, because shutting down expression of SG-associated proteins such as G3BP1 and TIA1 significantly compromises MxB’s efficiency in LINE-1 inhibition. Tests with mutated MxB indicated that nuclear membrane localization, oligomerization, and GTPase activity are all essential for MxB to suppress LINE-1. At last, the authors presented evidence suggesting that only some members of the Mx protein family are effective in LINE-1 regulation, and MxB can suppress other retrotransposons such as murine LINE-1, IAP, and MusD.

Taken together, this study suggested that changing the subcellular localization of ORF1p in host cells is the key mechanism for MxB-mediated LINE-1 suppression. And combining with previous reports, restricting ORF1p onto SG and/or p-bodies might be a common mechanism shared by various LINE-1 suppressors. Such information is both interesting and important for the LINE-1 community, and might shed light on future research in the Genetics and Genomics field. However, I am not fully convinced because some conclusions in this manuscript were not fully supported, while some phenomena were not fully investigated.

Major issues:

I first found it impressive that the authors tested MxB through three different systems: knockout, stable expression, and transient expression. And then I found myself confused because at most times it was difficult to find out which system the authors used for a statement in the text or a panel in a figure. For instance, the authors performed IP experiments in Fig. 2D, E, and F, but there was no description whether the experiments where performed in MxB stably expressed or transiently expressed HeLa cells. It would be nice if the authors clearly state which system was used when describing data in the text.

Some of the statements were not logical. In Line 121, the authors stated, “The association of MxB with LINE-1 RNA…”, while no LINE-1 RNA detection was performed with MxB interactants. The interaction between ORF1p and MxB cannot guarantee the binding between MxB and LINE-1 RNA. Because it is possible that, by binding ORF1p, MxB can sabotage the interaction between ORF1p and LINE-1 RNA. Besides, Fig. 2G needs improvement, as the ORF1p bands in the IP samples are hard to see, esp. when comparing to previous results in Fig. 2D, E, and F.

The use of Fig. S7 is also puzzling. Though it separates different Mx proteins from different species, the result had no impact on the selection of Mx proteins for subsequent LINE-1 retrotransposition assay. My best guess is that the authors chose some Mx proteins from each branch for subsequent tests. If true, the authors should elaborate the reason(s) in more details. Also, the procedures for the reconstruction of the phylogenetic tree was not mentioned. In addition, the tree should be bootstrap tested, with bootstrap values (>70) shown on the nodes of the tree to indicate the credibility. There is only one panel in Fig. S7, so the panel letter “A” is not necessary.

Some of the experiments were poorly designed and/or presented. For example, the authors should at least show one negative result for neomycin selection in Fig. S1B where the Neo-based LINE-1 retrotransposition assay is first presented. Such a negative result can be generated by transfecting HeLa cells with any retrotransposition-incompetent vectors (such as pJM111) reported by Moran et al. (Cell 1996), and can help to confirm LINE-1 retrotransposition events shown in Fig. S1B.

Why no elevation of ORF1p expression could be detected in cells transfected with LINE-1-EN- or LINE-1-EN+ in Fig. 3G?

Lines 149-154: The authors mentioned TIA1 here but no TIA1 images were shown in Fig. 4A and B.

Which experiment was conducted to generate Fig. 6E? And why some of the “relative mRNA levels” were not set as 1 for vector (control) groups?

Fig. S6B is not acceptable. The ORF1p band detected in the IP eluant of the negative control (i.e., the vector group) significantly compromises the interpretation of the whole IP results. The authors should repeat this experiment to achieve a better image as they did in Fig. 2D-F.

And the investigation for observed phenomena was not thorough in some cases. In Lines 100-108, the authors suggested that by suppressing LINE-1 retrotransposition, MxB has an impact on the expression of ORF1p. However, the authors completely neglected the possibility that MxB might induce the depletion of ORF1p, even when they later found that MxB can interact with ORF1p. Such an idea can be verified simply through Western blotting tests on cells transfected with ORF1p and MxB expressing vectors.

In Lines 165-171, the authors concluded that “MxB is located to G3BP1- and TIA1-bearing foci through interaction with ORF1p.” This also means the presence of ORF1p also altered the subcellular distribution of MxB which normally associates with nuclear membrane. The authors should test it to confirm, as such a phenomenon might interrupt other functions of MxB.

MxB suppressing murine SINE, IAP, and MusD is another example. Although it is impressive to known that MxB has a wider spectrum in regulating retroelements, these data introduce confusion to the study. To suppress LINE-1, MxB alters the subcellular localization of ORF1p. If ORF1p is an important component for the replication of murine SINE, IAP, and MusD, the authors should state it clearly, with proper references. If not, then the authors should investigate or at least speculate (with reasonable facts) whether MxB suppresses these retroelements through similar mechanisms.

In Discussion, the authors should provide possible reason(s) why the GTPase activity is essential for MxB-mediated LINE-1 suppression, or ORF1p re-localization.

Minor issues:

Text:

Full name for LINE-1 should be corrected to "long interspersed element type 1". No "nuclear" in the name. This convention dates back to early eighties (see Singer MF, Cell 1982). “L1” is also ok, which was used in the manuscript (e.g., L1-neo) but the authors failed to explain what “L1” strand for.

The authors used both “ORF1” and “ORF1p” to refer the open reading frame encoding the 40kDa protein. Be consistent. And please notice that “ORF1p” with the small p in most LINE-1-related literatures indicates the protein but not the ORF.

In Line 46, the authors stated, “LINE-1 is still active”, which can be misleading. Among ~500,000 LINE-1 copies in one single cell, only a very small fraction of them are competent in retrotransposition (this has been mentioned in many LINE-1 research papers and reviews). In addition, the word “active” is confusing, as many LINE-1 copies, though incompetent in replication, are active in transcription.

In Line 63, the authors stated, “digesting LINE-1 DNA by TREX-1”, while Li et al. reported that the exonuclease activity is not involved in TREX1-mediated LINE-1 suppression; instead, TREX1 inhibits LINE-1 by inducing the proteasome-mediated proteolysis of ORF1p (NAR 2017).

What did the authors mean by “LINE-1 DNA production” in Line 88?

Line 149: I believe the word “APOBC” here should be “APOBEC”.

The following statements need to be referenced:

Line 53: Not surprisingly,…and Alu insertion.

Line 70: HBV(ref) (I believe the authors meant to insert references here but forgot).

Figure:

Some of the figure legends are sloppy and confusing. For example, the description for Fig. 3G focused on tested HeLa cells, while the panel actually shows Western blotting results. The authors should correct such errors.

Fig. 2A, B, and C: Transfected vectors should be shown clearly by each set of images. Also, “LINE-1-MS2” in Fig. 2C is not a proper description, as green fluorescence can only indicate the localization of MS2-GFP, but not its status for LINE-1 RNA binding.

Fig. 2H and I: explain what the letters “C” and “N” stand for. And I suggest that the space between panel H and I should be increased. In current form, it is hard to determine whether the label “L1-Neo” belongs to H or I.

Bar charts in Fig. 2H, I, and J, and Fig. S6: the titles of y-axes should be changed, as it is not difficult to determine which protein has been targeted for nucleocytoplasm distribution.

Fig. 4A and B: was it MxB or MxB-EGFP used in these tests?

Fig. 6D: what is “huMaB” that was shown in the bar chart?

Explain those arrows used in Fig. 2A, 2B, 4A, 4B, S3, and S4.

In CMV-LINE-1-neoRT (the authors needs to confirm whether it is the pJM101 plasmid used in Moran et al. Cell 1996), there is a CMV promoter next to the 5’ side of LINE-1 5’-UTR. The authors missed it in Fig. S1A.

In the figure legend for S1B, it was stated that, “Number of neomycin-resistant colonies with the control vector is arbitrarily set as 1”. However, the y axes of bar charts in S1B (and other LINE-1 assay results) are showing numbers of colonies. And it should not be labeled as “colony assay” in the bar chart, which is a test but not a result. A proper title for the y-axes in bar charts showing neomycin-resistance-based L1 assay results can be found in Herrmann et al. Mol DNA 2018.

There is no description in the figure legend for Fig. S1D.

Fig. S4A: What is “MxBE”?

Explain the difference between Fig. S6D and the bar chart next to it (on the right)

Reviewer #4: The present study from Haung et al. investigates the anti-viral restriction factor MxB as also an inhibitor of LINE-1 retrotransposons. Some data reported here duplicates that of a previous report by Goodier et al. (2015, PLoS Genetics) including that MxB (but not MxA) decreases L1 retrotransposition, that MxB associates with L1 ORF1p in an RNA-dependent manner, and that the K131 MxB mutation which affects GTP binding restores retrotransposition. (These confirmatory data should be noted as such).

New data include demonstrating increase in L1 retrotransposition and endogenous L1 RNA and protein in an MxB KO cell line, decrease in L1 RNA and protein in a MxB knock-in cell line, and that MxB associates with the L1 RNP in cytoplasmic stress granules. Furthermore, the authors report that MxB reduces the amount of nuclear vs cytoplasmic ORF1p with a consequent reduction in H2AX-detected DNA damage, and that inhibition of stress granule components limits MxB inhibition of the L1. Thus, this study expands considerably on the previous limited work and proposes a mechanism for the effect of MxB on LINE-1s. As such it is worthy of consideration for publication in PLoS Genetics. However, while detailed supporting data has already been presented, I believe some additional controls and experiments are required to more strongly support the interesting claims made in this paper.

MAJOR:

There are no obvious specifically stated toxicity and cell growth rate assays for potential effects of MxB in transfected and knock-out cell lines. These should be shown. The N1-EGFP transfection control (p. 4, line 83) might be considered as an toxicity assay.

p. 4, line 111 (also abstract, p. 11, line 240, p. 12, line 248). The section title and conclusion, "MxB sequesters LINE-1 ORF1p in cytoplasmic bodies" is not necessarily correct. If this is claimed, the numbers of cells with ORF1p in SGs in the presence or absence of overexpressed MxB should be counted. It is stated on p. 13, line 272 that "we observed much greater LINE-1 ORF1p localization to G3BP1/MxB-positive cytoplasmic granules compared to the relatively more dispersed cytoplasmic distribution of ORF1p in the absence of MxB expression ((Fig. 4A, B and S4A, B)." If so this needs to be backed up by cell count quantification. In our experience overexpressed EGFP-ORF1p forms many stress granules in numbers of unstressed cell lines, including HeLa cells.

Furthermore, on p. 8, line 165 (also p. 14, line 291) it is noted that ectopic or endogenous MxB does not form arsenite-induced SGs in the absence of transfected ORF1p. That would suggest that overexpressed ORF1p sequesters MxB in SGs rather than the other way around. However, Fig. S3A, second row shows MxB-EGFP alone forming many SGs; furthermore, in the fourth row there are a significant number of ORF1p SGs that do not contain MxB protein -- so this is confusing regarding the authors' conclusion.

The definitive experiment is to see in the MxB KO cell line vs its wild-type control if overexpressed and endogenous ORF1p forms fewer stress granules/in fewer cells (both unstressed and arsenite-stressed). If the above claim is to be made, these experiments should be done. The monoclonal 4H1 anti-ORF1p antibody from Millipore is a good one; I believe it is available in China.

Fig. 3, F. This is unclear to me: are the controls and KO cell lines transfected with the L1? If not, is there any data for transfected cells?

Also, Fig. 3G, and p. 7, line 142. It is stated in the text that, "As a control, the endonuclease inactivated LINE-1 (EN-) did not increase γH2AX foci, and MxB overexpression had no effect (Fig. 3G)." No cell counts are presented in this figure to show the effect of the EN- control on γH2AX foci production: these should be shown. Also it is stated, "As a control, the endonuclease inactivated LINE-1 (EN-) did not increase γH2AX foci". However, there is a significant increase in γH2AX protein in both EN- and EN-transfected cells, which seems contradictory. Quantification of foci numbers could help in clarification.

p. 8, line 169. Also, it is concluded "Together, these data demonstrate that MxB associates with LINE-1 RNP in the stress granules, thus prevents nuclear import of LINE-1 proteins and diminishes LINE-1 retrotransposition." The authors are implying that MxB sequesters ORF1p in SGs to prevent further function. However, as noted above, to me the converse appears true, that ORF1p sequesters MxB.

Fig. 4D. The IP of G3BP1 by MxB is not entirely convincing due to background presence of MxB in the other two lanes. Also, there seems to be somewhat more G3BP1 in the MxB lane compared with vector. Can a better blot be shown? Quantification of band intensities might also help.

p. 9, line 186. There is a discrepancy with previous findings. In the present study, MxBdel25 lost the ability to inhibit retrotransposition while Goodier et al. reported that deleting the first 25 aa of MxB had no effect on retrotransposition (although their K131A mutation did, in agreement with the present study, a fact that should be also mentioned). This discrepancy should be noted and discussed. Perhaps epitope tags are having some effect.

MINOR.

p. 4, line 87. I point out that possible artifacts of quantitative PCR to measure levels of LINE-1 integrated cDNA in cells have been proposed in the literature, including generation of non-integrated ectopic cDNA by L1 RT which would then be amplified by PCR and misinterpreted as increased genomic insertions. I would suggest removing the word from '...to measure levels of LINE-1 integrated cDNA in cells".

Please note what arrows in all figures indicate. Some appear to show foci overlapping nuclei which is confusing.

Fig. S2D. The quality of the Western showing loss of MxB protein in KO cells is poor. Can a better one be presented?

Fig. 4E,F. Is the apparent increase in retrotransposition seen by knock-down of TIA-1 or G3BP1 significant? If so, this would appear worthy of comment in the text.

p. 7, line 144. It is noted, "These data further suggest that MxB inhibits the nuclear

import of LINE-1 RNP composed of LINE-1 RNA, ORF1p and ORF2p." While it is often assumed, I believe it has never been proven that ORF1p enters the nucleus bound to the L1 RNA. This caveat could perhaps be noted.

Fig. 6E, p. 11, line 223. It needs to be made clearer in the text that these mouse elements were detected by RT-PCR. Where were the primers located in the L1s?

p. 12, line 257. It is noted, "MxB inhibition of LINE-1 is CypA-independent (data not shown)". Show the data please if you wish to state this.

There are a number of quite minor English grammar errors scattered through the manuscript, a majority involving missing articles.

**Have all data underlying the figures and results presented in the manuscript been provided?**

Reviewer #1: Yes

Reviewer #2: Yes

Reviewer #3: Yes

Reviewer #4: Yes

PLOS authors have the option to publish the peer review history of their article (what does this mean?). If published, this will include your full peer review and any attached files.

Reviewer #1: **Yes: **Gilberto Betancor Quintana

Reviewer #2: **Yes: **Sandra Richardson

Reviewer #3: No

Reviewer #4: No

---

## [Decision Letter · Decision Letter 1]

14 Dec 2021

Dear Dr Guo,

Thank you very much for submitting your Research Article entitled 'MxB inhibits long interspersed element type 1 retrotransposition' to PLOS Genetics.

The manuscript was fully evaluated at the editorial level and by independent peer reviewers. The reviewers found the revised manuscript to be much improved and appreciated your detailed response to their concerns. Some reviewers still have lingering minor concerns which we feel can be addressed with modifications to the manuscript alone, without any additional experiments. When you address these comments, we will most likely be able to accept the manuscript without further input from the reviewers.

We therefore ask you to modify the manuscript according to the review recommendations. Your revisions should address the specific points made by each reviewer.

[LINK]

Yours sincerely,

Richard N McLaughlin, PhD

Guest Editor

PLOS Genetics

Gregory Barsh

Editor-in-Chief

PLOS Genetics

Reviewer's Responses to Questions

**Comments to the Authors:**

Reviewer #1: I find this version of the manuscript greatly improved. Authors have addressed most of my comments and results obtained are consistent. However, I still have some concerns about some microscopy images and with the explanation of some results:

Lines 40-41: the sentence “This is the first report attributing the restriction function of MxB to sequestering LINE-1 RNP, likely viral RNPs, to stress granules” is incorrect. Do authors mean like viral RNPs? If so, I think is still not correct, to the best knowledge of this reviewer, there is not evidence that cytoplasmic MxB is the one driving viral inhibition (certainly not in the case of HIV-1)

It would be advisable to include a blot showing MX2 in Ctr and KO lines in Fig 1D rather than going to sup Fig 2E. This would ease interpretation of the Figure

Sup Fig 3A. Have the authors quantified the colocalisation of ORF1p with GFP (in both, EGFP and MxB-EGFP)? Just by looking at the merge pictures is difficult to tell if colocalisation is higher with MxB-EGFP than with EGFP (Look at EGFP granules specially)

Line 151: “These results suggest that MxB associates with LINE-1 RNA and ORF1p” I am not convinced by this statement. Since there is no direct evidence of MxB interacting with LINE1 RNA, I think authors can’t rule out the diminish interaction in the presence of RNase due to a conformational change of ORF1p, different oligomerization status, etc, rather than because MxB directly binds the RNA

Line 163: should say ORF2p

Line 172: defective, not “detective”

Lines 163-174 and Fig 3: I am confused by this experiment. Cells depleted of MxB show an increase in γH2AX protein level and associated foci. Does this mean that MxB is a downmodulator of γH2AX expression and its absence results in γH2AX uncontrolled expression? Furthermore, the number of γH2AX foci in EGFP expressing cells transfected with L1-neo is comparable to the one seeing in MxB KO cells (specially clone 2). I think these results need further explanation

Line 197: an “in” is missing before agreement

Line 203: a “cell” is missing after HeLa

Line 206: It says “LNE” instead of LINE

Sup Fig 8B: why is the number of G418 resistant colonies reduced in the shCypA + vector condition compared to NC?

Line 223 onwards: I think it would help to notice somewhere in this paragraph that MxBdel25 is the same as the natural short isoform of MxB

Line 243: TNPO1 is transport receptor transportin 1, not transport “reporter”

Line 244: “which” should be removed

Fig 5: Shouldn’t MxBdel25+NLS be located inside the nucleus? In fig 5b it seems to localize just as MxA.

Sup Fig 9: localisation of MxBdel25 in panel A doesn’t recapitulate image shown in 5E. It seems as there was no signal for GFP and what is shown is background (including signal in the nucleus).

Lines 326-329: Discussion would be enriched if authors also mention the importance of the N-terminal domain of MxB for HIV-1 inhibition, involving capsid binding (Fricke et al., 2014), or regulation by phosphorylation (Betancor et al., 2021)

Line 371: remove ” at the end of line.

Lines 384-390: I think the authors should discuss the fact that MxB is able to inhibit retrotransposons in mice cells, which points to either MxB acting alone, or being assisted by human and murine proteins indistinctively.

Line 780: “show”, not showed

It’s my opinion figure legends should include what the error bars represent (SEM, SD…) or the statistical test used, to ease their interpretation.

Line 837: “non-human” instead of “no-human”

Line 843: “endogenous” instead of “endogenouse”

Line 844: “grey”, not “gray”.

Reviewer #2: Responses to major points:

1. Goodier et al and Doucet et al have now been appropriately cited.

2. The authors cited relevant literature to counter that ORF1p does not normally localize to the nucleus. They have repeated the immunofluorescence assay in Figure 2J using a smaller tag to detect ORF1p, and achieved a similar result to ORF1p-YFP.

3. The authors explain that the strong nuclear signal for MxB in the fractionation experiment in panels 2H and I is due to loading 5x more nuclear fraction than cytoplasmic fraction in their Western blot. They have put this information in the Materials and Methods, but I think it needs to be either in the main text or in the figure legend. I also think the explanation for how the normalization was done in panel 2I belongs in the figure legend, not the materials and methods. These experimental details are critical to interpreting the work.

4. I am still unclear on why sequestration of ORF1p and LINE-1 RNA in stress granules by MxB1 would lead to an over-all decrease in ORF1p and L1 RNA levels.

5. The figure legends are much more clear over all, and I appreciate that the Western blot in 3A has been repeated to show the relevant control bands.

6. I appreciate that it is difficult to detect endogenous ORF1p by immunofluorescence. I don't think it is critical to do so for the paper.

all minor points have been adequately addressed. I appreciate in particular the background on how MxA and MxB inhibit viruses.

Reviewer #3: By adding more data and explanations, the manuscript by Huang, et al. has been greatly improved for its logic, clarity, and significance. All my previous concerns have been addressed. However, with the ms revised, the authors also introduced some typos and gramma errors.

Here are some typo examples:

Line 94: RR261/261AA should be RR261/262AA

Line 206: LNE-1 should be LINE-1

Lines 242-251: Both TNPO-1 and TNPO1 were used in this paragraph. Be consistent.

Line 371: The quotation mark should be removed.

Line 416: The authors should check these cat.#: 66009-I-Ig and 67447-l-lg, as they were purchased from the same company and share similar numbering pattern but use different letters (the authors may set the font as “Times New Roman” for better display).

Line 470: I believe the “RNA free water” here should be “RNase free water”.

Lines 549-562: The authors should notice that some accession numbers listed in this paragraph were underlined.

Reviewer #4: Huang and group have made considerable efforts to improve the manuscript. While I am generally satisfied with the revised manuscript as it relates to my concerns, there are a couple of continuing points.

The authors now report to the reviewers, "We have scored the number of ORF1p-containing stress granules with and without MxB expression, observed no significant difference between the stable MxB-expressing cell line and control cell line" -- this new data should be noted in the Results section. They go on to note that they instead observed larger ORF1p-positive cytoplasmic foci. How was this enlargement of foci determined? I assume by visible inspection (rather than software analyses) -- this should be stated in the text and the number of cells examined to give this impression noted.

Furthermore on page 10 line 197 it is stated, " [In] agreement with these functional data, knockdown of endogenous G3BP1 or TIA-1 led to the loss of ORF1p/MxB-EGFP granules in the cytoplasm (Fig. S6C and S6F), further supporting the stress-granule nature of the ORF1p/MxB foci." If this statement is made, it needs to be quantitated with counts in the figure.

The authors comment, "This reviewer raised an important point, we have discussed this on page 17 line 359-365, “MxB is not an integral component of stress granules, since we did not observe MxB localization with stress granules that were induced with arsenite treatment (Fig. S7A and B). Expression of MxB itself did not cause stress granule formation either." However, this statement remains inconsistent with Fig. S3A, second row, which still shows MxB-EGFP alone forming many SGs. Does this mean endogenous MxB does not enter SGs, but that ectopic overexpression of MxB can induce SGs. I am willing to believe that, but these matters need to be clarified in the text.

Fig. 2A and B. White arrows are claimed to show ORF1p foci in the nuclei that disappear with MXB expression. While it has previously been reported that ORF1p enter nucleoli and can form foci in nuclei of some cells (Pereira et al. 2018 Mob. DNA), are the authors certain these foci are truly in the nucleus and not cytoplasmic foci imaged on top of nuclei? Z-stack analysis would confirm this and the point should be clarified in the Figure text.

Fig. 3G. The figure panel still shows the same Western as before for EN- and EN+. Please change to now show the WT and ORF1 mutant modified data.

In the Materials and Methods section the title "Quantification of integrated LINE-1 DNA by PCR" also needs to be changed to "Quantification of LINE-1 cDNA by PCR" as was done in the results section.

**Have all data underlying the figures and results presented in the manuscript been provided?**

Reviewer #1: Yes

Reviewer #2: Yes

Reviewer #3: Yes

Reviewer #4: Yes

PLOS authors have the option to publish the peer review history of their article (what does this mean?). If published, this will include your full peer review and any attached files.

Reviewer #1: **Yes: **Gilberto Betancor

Reviewer #2: No

Reviewer #3: No

Reviewer #4: No

---

## [Editor Report · Decision Letter 2]

12 Jan 2022

Dear Dr Guo,

We are pleased to inform you that your manuscript entitled "MxB inhibits long interspersed element type 1 retrotransposition" has been editorially accepted for publication in PLOS Genetics. Congratulations!

One point to note: In the previous reviews, I omitted the following statement from the comments from Reviewer 4, which I thought was a private comment to the editors. There is no action or response necessary on your part, but the reviewer has asked that we communicate this comment to you. My apologies this was not included in the previous comments:* "While I am happy to review the manuscript again, an editorial decision concerning these changes would be also acceptable. I think the revision of this manuscript perhaps has been especially difficult for the authors since they have had to address the concerns of four reviewers. In my opinion, that was at least one reviewer too many. Each reviewer requested additional work, which the authors have diligently attempted to satisfy in most cases. I feel this should be given consideration in the final decision." (Reviewer 4)*

Yours sincerely,

Richard N McLaughlin, PhD

Guest Editor

PLOS Genetics

Gregory Barsh

Editor-in-Chief

PLOS Genetics

Comments from the reviewers (if applicable):

**Data Deposition**

http://datadryad.org/submit?journalID=pgenetics&manu=PGENETICS-D-21-01057R2

**Press Queries**

---

## [Editor Report · Acceptance letter]

1 Feb 2022

PGENETICS-D-21-01057R2 

MxB inhibits long interspersed element type 1 retrotransposition 

Dear Dr Guo, 

We are pleased to inform you that your manuscript entitled "MxB inhibits long interspersed element type 1 retrotransposition" has been formally accepted for publication in PLOS Genetics! Your manuscript is now with our production department and you will be notified of the publication date in due course.

With kind regards,

Orsolya Voros

PLOS Genetics

On behalf of:
